# Yeast Eps15-like endocytic protein Pan1p regulates the interaction between endocytic vesicles, endosomes and the actin cytoskeleton

Junko Y Toshima[1,2], Eri Furuya[3], Makoto Nagano[2,3], Chisa Kanno[3], Yuta Sakamoto[3], Masashi Ebihara[3], Daria Elisabeth Siekhaus[4], Jiro Toshima[2,3]*

[1]Department of Liberal Arts, Tokyo University of Technology, Tokyo, Japan; [2]Research Center for RNA Science, Tokyo University of Science, Tokyo, Japan; [3]Department of Biological Science and Technology, Tokyo University of Science, Tokyo, Japan; [4]Institute of Science and Technology Austria, Klosterneuburg, Austria

**Abstract** The actin cytoskeleton plays important roles in the formation and internalization of endocytic vesicles. In yeast, endocytic vesicles move towards early endosomes along actin cables, however, the molecular machinery regulating interaction between endocytic vesicles and actin cables is poorly understood. The Eps15-like protein Pan1p plays a key role in actin-mediated endocytosis and is negatively regulated by Ark1 and Prk1 kinases. Here we show that *pan1* mutated to prevent phosphorylation at all 18 threonines, *pan1-18TA*, displayed almost the same endocytic defect as *ark1Δ prk1Δ* cells, and contained abnormal actin concentrations including several endocytic compartments. Early endosomes were highly localized in the actin concentrations and displayed movement along actin cables. The dephosphorylated form of Pan1p also caused stable associations between endocytic vesicles and actin cables, and between endocytic vesicles and endosomes. Thus Pan1 phosphorylation is part of a novel mechanism that regulates endocytic compartment interactions with each other and with actin cables.

*For correspondence: jtosiscb@rs. noda.tus.ac.jp

## Introduction

Endocytosis is the process by which cells internalize various molecules, such as proteins and lipids, from the plasma membrane and outside the cell. Recent live-cell imaging studies of yeast and mammalian cells have revealed that the actin cytoskeleton plays important roles in the formation and internalization of clathrin-coated vesicles (CCVs) and post-internalization events in the endocytic pathway, including vesicle transport and endosome motility (*Engqvist-Goldstein and Drubin, 2003*; *Girao et al., 2008*; *Kaksonen et al., 2006*). In the early stages of the endocytic pathway, transient actin polymerization at the endocytic site is required for the formation and internalization of a CCV (*Kaksonen et al., 2003*; *Merrifield et al., 2002*). This process is regulated by several actin nucleation promoting factors (NPFs), including the type I myosins Myo3/5p, the actin binding protein Abp1p, the yeast WASP homologue Las17p, and the Eps15-like protein Pan1p (*Goode et al., 2015*; *Weinberg and Drubin, 2012*). Compared to Las17p and Myo3/5p, Pan1p has a lower NPF activity, and mutation in the Arp2/3 complex binding region of Pan1p causes only a minor defect in actin polymerization at endocytic sites because of the functional redundancy with Las17p (*Sun et al., 2006*; *Toshima et al., 2005*). However, a form of Pan1p mutated to prevent phosphorylation at 15 threonines, Pan1-15TA, directly binds to F-actin with high affinity and its expression causes abnormal cytoplasmic actin concentrations, also called actin clumps (*Toshima et al., 2005*). Thus, Pan1p could

**eLife digest** The cells of all eukaryotes – including plants, animals and fungi – absorb many substances that they need from their surroundings by forming pockets around them, and then pinching off these pockets to create structures called vesicles. Clathrin is a protein that acts as a scaffold for these vesicles.

Inside a eukaryotic cell, clathrin-coated vesicles first go to a structure known as an endosome, possibly by following a track made from filaments of a protein called actin. Researchers have shown previously that a yeast protein called Pan1 binds to actin filaments and helps the cells to create clathrin-coated vesicles. However it was unclear if the Pan1 protein is also important for transporting clathrin-coated vesicles to endosomes.

Previous studies have also shown that adding phosphate groups on to the Pan1 protein prevents it from binding to clathrin-coated vesicles or actin filaments. Now, Toshima et al. show that a mutant version of the Pan1 protein, which cannot be modified in this way, can bind stably to both clathrin-coated vesicles and the actin filaments and connect them together. The experiments also showed that, in yeast cells that only produce the mutant version of Pan1, clathrin-coated vesicles bind stably to endosomes without the need for actin. Thus, these findings show that the addition of phosphate groups onto Pan1 is part of a mechanism that regulates the interactions between clathrin-coated vesicles, endosomes and actin filaments.

Following on from this work, one future challenge is to find which proteins directly connect clathrin-coated vesicles with endosomes. It will also be important to investigate if similar mechanisms are used in the cells of mammals.

be acting after F-actin is assembled by other NPFs as well as or instead of during the initiation of actin polymerization during endocytosis.

Pan1p's abilities to bind F-actin and promote actin polymerization are regulated by the Prk1 family protein kinases Prk1p and Ark1p, which are related to the mammalian proteins GAK and AAK1 (*Smythe and Ayscough, 2003*). The Prk1 family kinases are important regulators of endocytosis and the actin cytoskeleton in both yeast and mammalian cells (*Smythe and Ayscough, 2003*). In budding yeast, Ark1p and Prk1p are recruited to endocytic sites 1–2 s after commencement of actin assembly and CCV internalization, and phosphorylate several endocytic proteins, including Sla1p, Ent1/2p, Yap1801/2p, Scd5p, and Pan1p, to disassemble endocytic coat proteins and actin (*Cope et al., 1999*; *Henry et al., 2003*; *Toret et al., 2008*; *Watson et al., 2001*; *Zeng and Cai, 1999*; *Zeng et al., 2001*). Pan1p is one of the key targets of Ark1/Prk1 kinases, and phosphorylation of Pan1p by Ark1/Prk1 kinases is believed to be important for disassembly of the Pan1p complex, composed of several endocytic proteins (*Toshima et al., 2005*; *Wendland and Emr, 1998*; *Zeng and Cai, 1999*; *Zeng et al., 2001*). Interestingly, disruption of the normal phosphorylation cycle by deletion or chemical inhibition of Ark1/Prk1 kinases leads to the concentration of actin in association with endocytic vesicles (*Sekiya-Kawasaki et al., 2003*; *Toshima et al., 2005*), suggesting a role for Pan1p and other substrates in regulating interaction between endocytic vesicles and the actin cytoskeleton.

After being internalized, endocytic vesicles move away from the plasma membrane in an association with actin cables that is still mechanistically unexplained (*Huckaba et al., 2004*; *Toshima et al., 2006*). Yeast actin cables, which are bundles of actin filaments that align along the long axis of budding yeast, are crucial for the establishment of cell polarity (*Yang and Pon, 2002*). Actin cables are also used as tracks for polarized transport during the secretion of exocytic vesicles and the segregation of organelles from mother to daughter cells (*Bretscher, 2003*). Many of these types of transport along actin cables are known to depend on the type V myosins, Myo2/4p, which mediate the movement of cargo from the minus to plus ends of actin filaments (*Bretscher, 2003*). However, transport of endocytic vesicles along actin cables is not likely to depend on these myosins, because a temperature sensitive mutant of *MYO2 (myo2-66)* or a deletion of *MYO4* gene did not exhibit any defect in endocytosis (*Govindan et al., 1995*; *Haarer et al., 1994*). Other myosins, such as type II myosin (Myo1p) and type I myosin (Myo3/5p) also do not seem to mediate this transport. Myo1p has an

important role in controlling actin cable dynamics at the bud sites or neck, but it is not localized to endocytic vesicles (*Huckaba et al., 2006*). Myo3/5p are necessary for promoting actin assembly and endocytosis at cortical patches, but they stay at the cell cortex when endocytic vesicles are internalized along actin cables (*Sun et al., 2006*). Interestingly, a previous study demonstrated that endocytic vesicle movement occurs at the same velocity and in the same direction as the movement of actin cables (*Huckaba et al., 2004*). They also reported that an endocytic vesicle stays at the same position on the cable and moves together with the actin cable, suggesting that endocytic vesicles are fixed on the actin cables and move as a result of actin cable flow (*Huckaba et al., 2004*). In addition to endocytic vesicles, early endosomes also associate with the actin cytoskeleton, and the motility of endosomes is significantly inhibited by treatment of latrunculin A (LatA), a drug that sequesters actin monomers (*Chang et al., 2003*; *Fernandez-Borja et al., 2005*; *Toshima et al., 2006*; *Voigt et al., 2005*). Similarly to endocytic vesicles, early endosome motility also does not depend on Myo2/4p (*Toshima et al., 2006*). These results suggest that unknown molecular mechanisms exist that bind endocytic vesicles and endosomes to actin cables.

We sought to understand the role of Pan1 phosphorylation during endocytosis using a form of Pan1 that mimics the *ark1Δ prk1Δ* phenotype. We examined cells expressing Pan1-18TA, which is mutated to prevent phosphorylation at all 18 threonines; this mutant showed almost the same endocytic defect as *ark1Δ prk1Δ* cells, resulting in stable association between endocytic vesicles and actin cables. Interestingly, the *pan1-18TA* mutant also leads to accumulation of early endosomes in actin clumps. Thus, phosphorylation of Pan1p seems to regulate the interaction between endocytic compartments and the actin cytoskeleton.

## Results

### Pan1p is the major in vivo target of Ark1/Prk1 kinases during their regulation of endocytosis

Our group had previously demonstrated that expression of a form of Pan1 containing a mutation of 15 Ark1p/Prk1p consensus sequences (LxxQxTG) to alanine causes an endocytic defect and abnormal clumping of actin in the cytosol. However, the defect in the *pan1-15TA* mutant was not as pronounced as that in the *ark1Δ prk1Δ* mutant (*Toshima et al., 2005*). We first sought to determine if the presence of other functionally important phosphorylation sites in Pan1p was responsible for the difference in phenotypes. In a previous intensive investigation, Cai and colleagues identified the [L/I/V/M]xx[Q/N/T/S]xTG motif as a further potential site of phosphorylation by Ark1/Prk1 kinases (*Huang et al., 2003*). Pan1p contains three more such Ark1p/Prk1p consensus sequences (MQPNIT$^{464}$G, MMPQTT$^{480}$G, and MMPQTT$^{487}$G) all located in the second LR region (*Figure 1A*) (*Huang et al., 2003*). When we additionally mutated these sites to create *pan1-18TA*, we observed a more severe growth retardation phenotype than in *pan1-15TA* (*Figure 1B*). The Pan1-18TA protein was expressed normally, but its phosphorylation was mostly inhibited (*Figure 1C*). *pan1-18TA* mutant cells displayed prominent actin concentrations and a more severe defect in endocytic internalization (*Figure 1D,E*). Pan1-18TA-GFP also showed defects in localization, with 95% colocalizing with actin clumps or smaller, peripheral actin patches, similar to Pan1-15TA (*Figure 1F*) (*Toshima et al., 2005*). This is in contrast to wild-type cells, in which Pan1p is recruited to cortical patches early, arriving ~20 s before actin is detected, and associates with actin for ~10–15 s (*Kaksonen, 2003*), resulting in ~30% of Pan1p colocalizing with Abp1p (*Figure 1F*).

We next entertained the hypothesis that Prk1p phosphorylation of some of its other known targets such as Sla1p, Ent1/2p, Yap1801/2p and Scd5p (*Watson et al., 2001*; *Zeng et al., 2001*; *2007*), might also play a role in regulating actin organization and endocytosis. These target proteins were shown to be phosphorylated by Prk1p in vitro, but significant phenotypes caused by mutations of their phosphorylation-sites have not been observed (*Henry et al., 2003*; *Huang et al., 2003*; *Watson et al., 2001*; *Zeng et al., 2001*). Sla1p contains the most potential Prk1 phosphorylation sites among these proteins (*Zeng et al., 2001*). We therefore mutated the threonines in all 10 of these [L/I/V/M]xx[Q/N/T/S]xTG sites (*Huang et al., 2003*) to alanine (*sla1-10TA*), integrated this mutant into the endogenous *SLA1* locus, and analyzed the phenotypes (*Figure 1—figure supplement 1A*). We first confirmed that the Sla1-10TA mutant was expressed at similar levels to the wild-type protein, and that its phosphorylation was severely inhibited (*Figure 1—figure supplement 1B*).

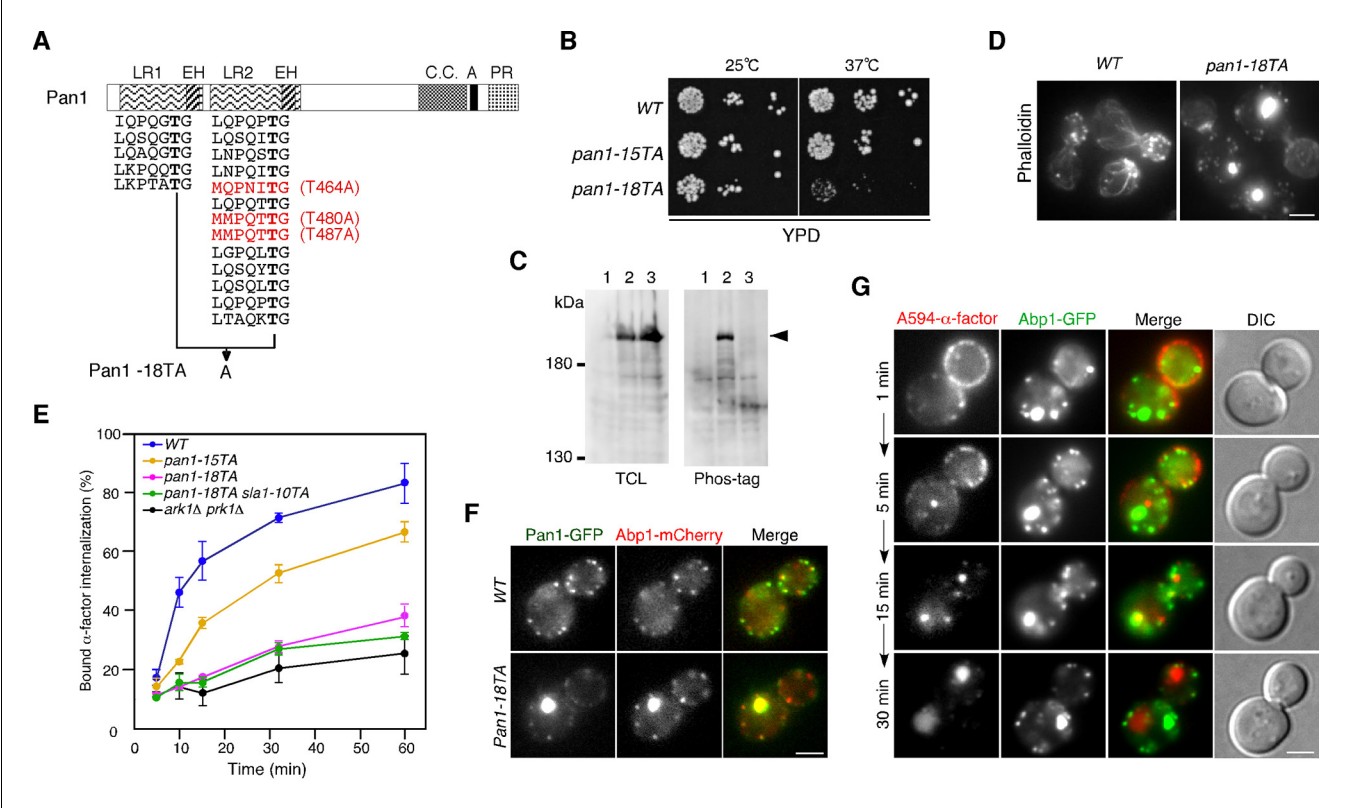

**Figure 1.** Construction and characterization of a Pan1p phosphorylation-site mutant. (A) Structure of a Pan1p phosphorylation mutant. The two amino-terminal Eps15 homology (EH) domains, long repeat (LR) regions, predicted coiled-coil (CC), acidic region (A), and carboxy-terminal prolin-rich domain (PR) domain, are indicated. The fifteen consensus phosphorylation sites previously mutated in Pan1-15TA are indicated below the protein in black. The three additional sites mutated in Pan1-18TA are in red. (B) Plate showing the growth phenotype of *pan1-15TA* and *pan1-18TA* mutants. A dilution series of cells was plated on YPD plates and incubated for 2–3 days at 25 or 37°C, respectively. (C) Analysis of the phosphorylation state of the Pan1-18TA mutant. Protein expression was analyzed by immunoblotting 20 µg of total cell lysate (TCL) with an anti-GFP antibody (left panel). Phosphorylated proteins were purified from TCL with Phos-tag agarose, run on SDS-PAGE and immunoblotted with the anti-GFP antibody (right panel) as described in Materials and methods. Lane 1, JJTY369; lane 2, JJTY509; lane 3, JJTY5486. (D) Alexa Fluor 488-phalloidin staining of fixed wild-type and *pan1-18TA* cells to visualize actin. (E) The effect of Pan1p phosphorylation-site mutations on endocytic internalization. Radiolabeled α-factor internalization assays performed on wild-type (blue), *pan1-15TA* (yellow), *pan1-18TA* (magenta), *pan1-18TA sla1-10TA* (green), or *ark1Δ prk1Δ* (black) cells at 25°C. Each curve represents the average of three independent experiments, and error bars indicate the SD at each time point. (F) The localization of Pan1-GFP in wild-type and *pan1-18TA* cells. Cells expressing Pan1-GFP and Abp1-mCherry were grown to early to mid-logarithmic phase in YPD medium at 25°C and observed by fluorescence microscopy. Merged images of GFP and mCherry channels are shown in the right panels. (G) Endocytic cargo is transported to the vacuole through the actin clumps in *pan1-18TA*. *pan1-18TA* cells were labeled with A594-α-factor as described in the Methods. The images were acquired simultaneously at 1, 5, 15, and 30 min after washing out unbound A594-α-factor and warming the cells to 25°C. Scale bars, 2.5 µm.

The following figure supplement is available for figure 1:

**Figure supplement 1.** Construction and characterization of a Sla1p phosphorylation-site mutant.

While cells lacking the *SLA1* gene were temperature-sensitive for growth at 37°C, the *sla1-10TA* mutant exhibited almost the same growth as wild-type cells (*Figure 1—figure supplement 1C*). Alexa Fluor 488-phalloidin staining of F-actin in fixed *sla1-10TA* mutants closely resembled that of wild-type cells, with brightly stained actin patches and weakly stained actin cables (*Figure 1—figure supplement 1D*). We next examined the dynamics of the clathrin-coat module and the actin patches in the *sla1-10TA* mutants, using GFP-tagged Sla1-10TA and Abp1-mCherry as markers respectively (*Kaksonen et al., 2003*). Consistent with previous reports, Sla1p and Abp1p patches formed in the cell cortex with lifetimes of 36 ± 7 s and 13 ± 3 s, respectively, culminating in inward movement (*Figure 1—figure supplement 1E,F*) (*Kaksonen et al., 2003*). Sla1-GFP localization was immediately

followed by a burst of Abp1-mCherry recruitment in wild-type cells (*Figure 1—figure supplement 1E*). In the *sla1-10TA* mutants, Sla1p and Abp1p patches formed and disappeared with the typical inward movement, and their lifetimes were slightly prolonged to 43 ± 11 s and 15 ± 3 s, respectively (*Figure 1—figure supplement 1F*). We also examined the effect of *sla1-10TA* mutants on endocytic internalization by assessing the ingression of $^{35}$S-labeled α-factor, and found that it was only slightly affected (*Figure 1—figure supplement 1G*). Furthermore, the *pan1-18TA sla1-10TA* double mutant exhibited only a negligible additive effect, when compared to the *pan1-18TA* single mutant (*Figure 1E*, and *Figure 1—figure supplement 1D,H*). These findings indicate that Pan1p is the major in vivosubstrate of Ark1/Prk1 kinases during their regulation of endocytosis.

We next visualized sequential steps in the endocytic pathway in the *pan1-18TA* mutant using Alexa Fluor 594-labeled yeast mating pheromone α-factor (A594-α-factor), a marker of endocytosis (*Toshima et al., 2006*). Interestingly, internalized A594-α-factor moved to actin clumps in *pan1-18TA* before being transported to the vacuole, while the actin remained in clumps (*Figure 1G*). This result suggests that the endocytic cargo can transit through the actin clumps before arriving at the vacuole.

## Early endosomes are localized at actin clumps in the *pan1-18TA* mutant

We next sought to determine if other organelles along the endocytic route also accumulate in these actin clumps. To this end, we employed Vps21p and Sec4p as markers of Rab proteins that function in the endocytic or exocytic pathway (*Hutagalung and Novick, 2011*), Vps8p and Vps11p as markers of the CORVET and HOPS complexes (*Balderhaar and Ungermann, 2013*), a set of proteins from the ESCRT complex (Hse1p, Mvb12p, Vps36p, and Vps24p) (*Bilodeau et al., 2002*; *Hurley, 2008*), Vps4p, Ear1p, and Vps15p as markers of MVBs (Ear1p, and Vps15p) (*Burda et al., 2002*; *Leon et al., 2008*), Vps26p as a marker of the retromer complex (*Seaman, 2004*), Vps52p as a marker of the GARP (Golgi-associated retrograde protein) complex (*Bonifacino and Hierro, 2011*), and Sec7p as a marker of the *trans*-Golgi network (*Franzusoff et al., 1991*). None of these proteins show clump-like localization in wild-type cells (*Figure 2—figure supplement 1*). Among the 14 proteins examined, three – Hse1p, Mvb12p, and Vps36p – showed a clear change in localization to Abp1-mCherry-labeled actin clumps in the *pan1-18TA* mutant (*Figure 2A,B*). Hse1p, Mvb12p, and Vps36p, which function at an early stage of the ESCRT pathway on the way to the MVB, were contained in 60–65% of the actin clumps, whereas Vps24p and Vps4p, which function at a later stage of the ESCRT pathway, were contained in 25–30% of the actin clumps (*Figure 2A,B*). Vps8p and Vps11p, which mediate early to late transitions of endosomes, exhibited levels of actin clump localization (~30%) similar to that of Vps24p (*Figure 2B*). In contrast, Vps26p or Vps52p, both of which are required for retrograde transport from late endosomes to the Golgi, showed lower localization (~15% and ~7%, respectively), and Sec7p and Sec4p, which reside on the Golgi or secretory pathway, were rarely contained in actin clumps in the *pan1-18TA* mutant (<10%) (*Figure 2A,B*). These results indicate that earlier stage endosomes are highly localized to actin clumps in *pan1-18TA* mutant cells.

We wished to confirm that early endosomes accumulate at actin clumps in *pan1-18TA* mutant cells. As we had seen the ESCRT-0 component Hse1p localize to actin clumps in the mutant, we wished to investigate its organellar localization in wild-type cells more precisely; therefore we tagged Hse1p with three tandem repeats of GFP (3GFP) or mCherry and confirmed their functionality (*Figure 3—figure supplement 1A*). Hse1-3GFP was clearly detected as numerous small puncta throughout the cytoplasm and prevacuolar compartments (PVCs) (*Figure 3—figure supplement 1B*). Examining Hse1-3GFP in the overlay of 30 consecutive time-lapse frames made it easy to distinguish the Hse1p localizing at endosomes in the cytoplasm (*Figure 3—figure supplement 1B*, right cell) and at the PVCs (*Figure 3—figure supplement 1B*, left cell). By comparing the localization of Hse1-mCherry with GFP-tagged markers, we found that Hse1p exhibited high colocalization with Mvb12p, partial colocalization with Ear1p and Vps26p, and little colocalization with Sec7p (*Figure 3—figure supplement 1C,D*). We next utilized A594-α-factor to compare the spatiotemporal localization of Hse1p with Vps26p in the endocytic pathway (*Figure 3—figure supplement 1E,F*) (*Toshima et al., 2006*; *2009*). Hse1-3GFP was highly colocalized with A594-α-factor-labeled endosomes at 5–10 min after α-factor internalization, whereas Vps26-GFP was mainly colocalized with A594-α-factor-labeled endosomes at 10 min (*Figure 3—figure supplement 1G*). In *pan1-18TA* cells, similar levels of Hse1p also colocalized with Vps26p (*Figure 3—figure supplement 1D,H*). These

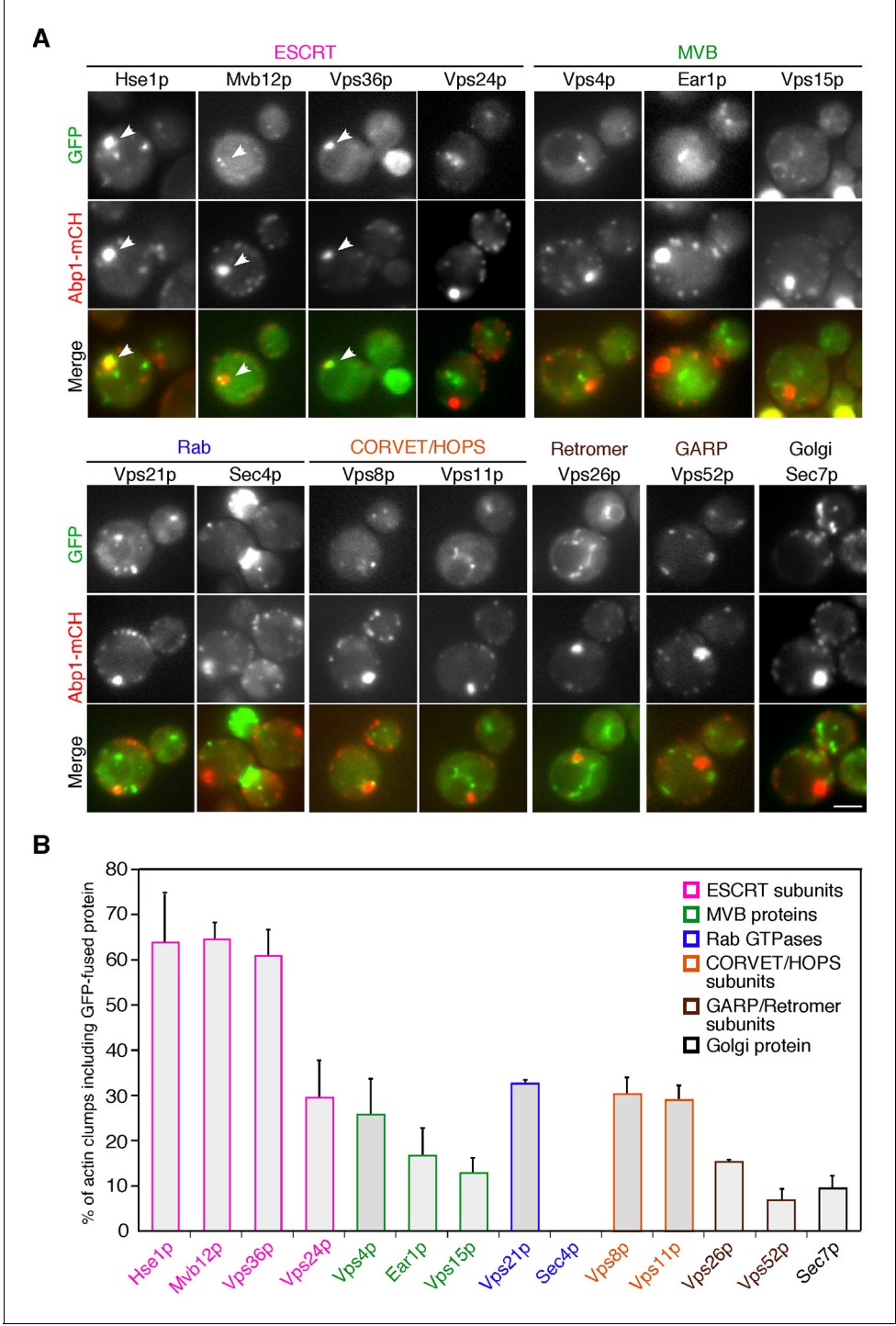

**Figure 2.** The localization of endosomal proteins in *pan1-18TA* cells. (**A**) Localization of GFP-tagged endosomal proteins in *pan1-18TA*. *pan1-18TA* cells expressing Abp1-mCherry and GFP-tagged endosomal proteins were grown to early to mid-logarithmic phase in YPD medium at 25°C and observed by fluorescence microscopy. Merged images of GFP and mCherry channels are shown in the lower panel. Arrowheads indicate examples of colocalization. Scale bar, 2.5 µm. (**B**) Quantification of actin clumps including GFP-tagged endosomal proteins. The percentages were calculated as the ratio of actin clumps (*n* = 100) colocalizing with each protein in each experiment. Error bars indicate the standard deviation (SD) from at least three independent experiments.

The following figure supplement is available for figure 2:

**Figure supplement 1.** Localization of endosomal proteins in wild-type cells.

data indicate that Hse1p is widely localized from early to late endosomes and partially colocalized with Vps26p at late endosomes both in wild-type and *pan1-18TA* cells. Unexpectedly, Vps21p, yeast Rab5, which is known to be localized at early-to-late endosomes (*Cabrera et al., 2013*; *Puchner et al., 2013*; *Toshima et al., 2014*), exhibited lower localization at the actin clumps (~33%), compared to Hse1p (*Figure 2A,B*). To determine the relative localization of Hse1p and Vps21p at early stage of endocytosis, we slowed endocytic transport by removing glucose from culture medium (*Aoh et al., 2011*), and compared their localization with internalized A594-α-factor. Consistent with a recent study by Arlt et al. reporting that an ESCRT-I subunit, Vps23, is recruited to endosomes earlier than Vps21p (*Arlt et al., 2015*), we found that Hse1p colocalizes with A594-α-factor slightly more than Vps21p at 10 min after α-factor internalization (*Figure 3—figure supplement 2A*). These results, therefore, suggest that endosomes at the early stage of endocytosis are highly localized to actin clumps.

## Actin-dependent motility of Hse1p-residing endosomes

The localization of early endosomal proteins at actin clumps in the *pan1-18TA* mutant suggests that early endosomes might associate with the actin cytoskeleton in wild-type cells. Previous studies have also indicated that actin cables mediate the directed movements of early endosomes (*Chang et al., 2003*; *Toshima et al., 2006*), but which endosomes, and how they associate with actin cables, has not yet been clarified. We utilized Hse1p as a marker to address these questions. However because Hse1p is found at early to late endosomes, we classified these endosomes into two categories using Vps26-mCherry: endosomes not labeled with Vps26-mCherry (early stage endosomes) and endosomes labeled with Vps26-mCherry (late stage endosome) (*Figure 3A,C*). Vps26-mCherry mostly colocalized with Hse1p at the vacuolar membrane (*Figure 3A,C*). Quantification of endosome velocity revealed that early-stage endosomes moved with an average speed of 125 ± 119 nm/s (*n* = 100), whereas late-stage endosomes moved with an average speed of 156 ± 130 nm/s (*n* = 100) in wild-type cells (*Figure 3A,B*). We then investigated the effects of the actin-sequestering drug, LatrunculinA (LatA), on the movement of these endosomes. Concomitantly, LatA treatment led to a significant decrease in the velocity of early-stage endosomes (~23 ± 47 nm/s) (*Figure 3C,D*). In contrast, the velocity of late-stage endosomes was not significantly affected by LatA treatment (~119 ± 112 nm/s) (*Figure 3C,D*). Similar results were obtained by analyzing Vps21p-containing endosomes (*Figure 3—figure supplement 2B*). The velocity of Vps21p-containing endosomes not labeled with Vps26-mCherry (~146 ± 128 nm/s) was decreased by LatA treatment (~55 ± 42 nm/s), whereas that of ones labeled with Vps26-mCherry (~138 ± 128 nm/s) was not significantly affected (~99 ± 91 nm/s). To further confirm the association between Hse1p-labeled endosomes and the actin cytoskeleton, we labeled actin cables with tdTomato-tagged Abp140p (*Yang and Pon, 2002*) in wild-type cells. Simultaneous imaging revealed that Hse1p-labeled endosomes localized along, and moved on, actin cables (*Figure 3E,F*, and *Video 1*). Since two distinct sets of actin assembly-promoting machinery have been identified in yeast, the Arp2/3 complex and formins (*Goode et al., 2015*), we next utilized specific inhibitors toward these regulators. The Arp2/3 complex inhibitor CK-666 specifically disassembled Arp2/3 complex-dependent actin patches, whereas SMIFH2 disassembled formin-dependent actin cables (*Nolen et al., 2009*; *Rizvi et al., 2009*). As expected, SMIFH2 treatment led to actin cable disassembly, followed by a decrease in the velocity of endosomes (~60 ± 53 nm/s). In contrast, CK-666 inhibited vesicle internalization, but endosome motilities were little affected (~126 ± 123 nm/s) (*Figure 3G,H*, and *Video 2*). These results indicate that the movement of early-stage endosomes is dependent on formin-dependent actin polymerization.

## Pan1p-labeled endocytic vesicles associate with actin cables in the *pan1-18TA* mutant

In addition to endosomes, endocytic vesicles are also known to associate with actin cables (*Huckaba et al., 2004*; *Toshima et al., 2006*), but the molecules that regulate this association remain unclear. Therefore, we examined the effect of the *pan1-18TA* mutation on the localization and dynamics of actin cables and endocytic vesicles. In wild-type cells, actin cables are highly dynamic polarized structures (*Figure 4—figure supplement 1A* and *Video 3*) (*Yang and Pon, 2002*). In contrast, the *pan1-18TA* mutant exhibited less polarized and more aggregated actin cable structures, some of which associated with the large actin concentrations (*Figure 4—figure*

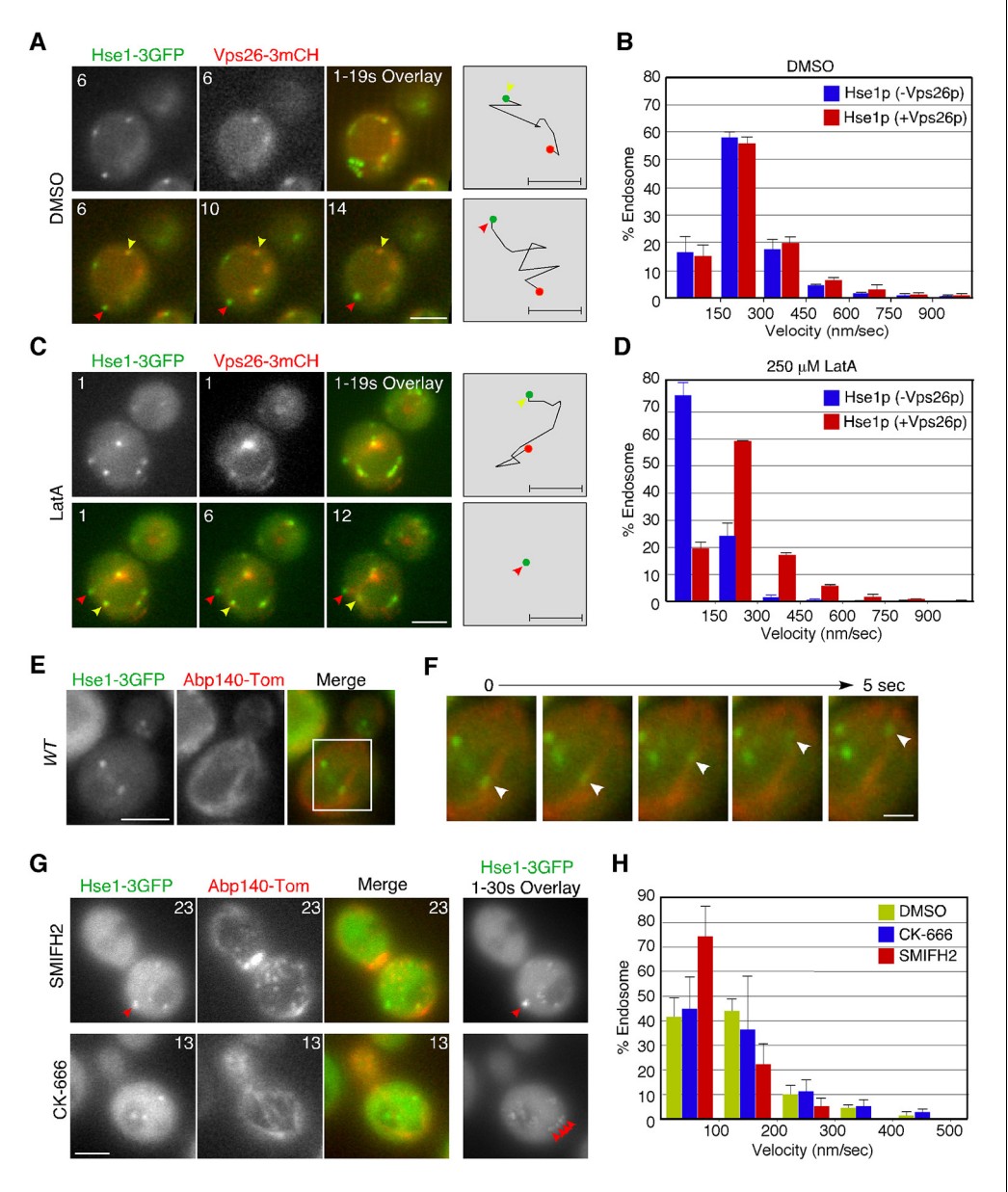

**Figure 3.** Interaction of endosomes expressing Hse1-3GFP with the actin cytoskeleton. (**A** and **C**) Movement of Hse1-3GFP-containing endosomes in living cells. Wild-type cells expressing Vps26-3mCherry and Hse1-3GFP were grown to log phase at 25°C, treated with DMSO (LatA-) (**A**) or 250 µM LatA (**C**) for 30 min at 25°C, and subsequently imaged at 1 s intervals. In the upper row the left two panels show the individual channels and the right most image shows a merged overlay of the signal from both channels in the first 19 s. The lower row shows merged images of both channels at indicated time. Yellow or red arrowheads indicate examples of Hse1p-containing endosomes that do or do not co-label with Vps26p, respectively. Scale bars, 2.5 µm. The schematic panels on the right show tracking of the endosome indicated by yellow or red arrowheads in the lower microscope images. Scale bars, 0.5 µm. (**B** and **D**) Quantification of the velocity of Hse1p-containing endosomes. Endosome velocities were acquired at 1 s intervals and categorized according to a velocity range. (**E**) Hse1p-containing endosomes move along actin cables. Wild-type cells expressing Hse1-3GFP and Abp140-tdTomato were grown to early to mid-logarithmic phase and each image pair was acquired simultaneously at 1 s intervals. Scale bar, 2.5 µm. (**F**) Higher magnification view of the boxed area in (**E**) at successively later time points specified in s above. Arrowheads indicate an endosome moving along an actin cable. (**G**) The effect of SMIFH2 or CK-666 on movement of Hse1-3GFP-containing endosomes. Wild-type cells expressing Abp140-tdTomato and Hse1-3GFP were grown to log phase at 25°C, treated with 25 µM SMIFH2 (the upper row) or 100 µM CK-666 (the lower row) for 30 min at 25°C, and subsequently imaged at 1 s intervals. The left three panels show the individual channels

*Figure 3 continued on next page*

*Figure 3 continued*

and their merged image for a particular time point, and the right most image shows a merged overlay of Hse1-GFP signals in 30 sec. Red arrowheads indicate examples of Hse1p-containing endosomes. Scale bars, 2.5 µm. (**H**) Quantification of the velocity of Hse1p-containing endosomes. Endosome velocities were acquired at 1 s intervals and categorized according to a velocity range.

The following figure supplements are available for figure 3:

**Figure supplement 1.** Localization of 3GFP- or mCherry-tagged Hse1p in wild-type cells.

**Figure supplement 2.** Localization and movement of GFP-Vps21-containing endosomes in living cells.

---

*supplement 1A* and *Video 3*). The movement of this Pan1p/actin aggregate in *pan1-18TA* was sensitive to SMIFH2 but not CK-666, suggesting that the aberrant structure associates with actin cables (*Figure 4—figure supplement 1B–E*). In a recent study, we showed that ~85% of endocytic vesicles were internalized along actin cables at the internalization step of endocytosis (*Toshima et al., 2015*). Here we show that in this step, Pan1-mCherry-labeled vesicles in wild-type cells associated with actin cables for ~4.6 s and moved on cables about 0.4 µm after internalization (*Figure 4A,B,C*). This association and these movements were significantly decreased by treating with 25 µM SMIFH2 (*Figure 4B,C,D*). Over 80% of Pan1-mCherry-labeled endocytic vesicles, even in single focal plane images, also associated with and internalized along actin cables in the *pan1-18TA* mutant (*Figure 4E*). In contrast to wild-type cells, Pan1-18TA-mCherry structures became stably associated with peripheral patches, as well as actin clumps, labeled by Abp140-3GFP (*Figure 4E*). Interestingly, live-cell imaging revealed that many of internalized patches labeled by Pan1-18TA-mCherry stably associated with actin cables over 30 s, and move on actin cables more than 3.0 µm (*Figure 4B,C,E*, and *Video 4*). We also wished to determine the relationship of endocytic vesicles and endosomes in the absence of Pan1 phosphorylation. In the *pan1-18TA* mutant, we often observed that peripheral Pan1-18TA-mCherry patches colocalized and moved together with Hse1p-labeled endosomes (*Figure 4F,G*, and *Video 5*), whereas such colocalization was rarely observed in wild-type cells (*Figure 4F*). These findings suggest that in the *pan1-18TA* mutant, endocytic vesicles and endosomes interact before fusion while they are both associated with actin cables, potentially tethering them together. Our data was collected with a wide field microscope; high-speed confocal microscopy could improve the quality of our results as it would permit 3-D analysis (*Kurokawa et al., 2013*). However, since some vesicles and endosomes move in a single focal plane, the simultaneous double color live cell imaging used in this study permits analysis leading to substantive biological insights.

## Interaction between endocytic vesicles and actin cables upon inhibition of Prk1p

To further investigate the phosphorylation-dependent association between endocytic vesicle and actin cable, we next used an analogue-sensitive mutant of Prk1p in cells lacking Ark1p (*ark1△ prk1-as3*) (*Sekiya-Kawasaki et al., 2003*). This mutant shows specific sensitivity to 1NA-PP1, an ATP analogue, and enabled us to investigate the direct and immediate consequence of Prk1p inactivation for the association between endocytic vesicles and actin cables. In the mutant untreated with 1NA-PP1, Pan1-mCherry-labeled vesicles only transiently associated with actin cables, similar to wild-type cells (*Figure 5A* and *Video 6*). However, at 1 min after treatment of the mutant with 100 µM 1NA-PP1, endocytic vesicles stably associated with, and moved on actin cables. By 3 min small aggregates containing Pan1p that associated with actin cables were formed (*Figure 5A* and *Video 6*). At 10 min after 1NA-PP1 treatment, a large actin clump that stably associated with actin cables was formed in the mutant, similar to the *pan1-18TA* mutant (*Figure 5A* and *Video 6*; also see *Figure 4C*). These observations support the idea that Pan1p phosphorylation by Ark1/Prk1 kinases is necessary for the rapid dissociation of endocytic vesicles from actin cables.

Pan1p can bind directly to F-actin and its binding activity is regulated by phosphorylation through the Ark1/Prk1 kinases (*Toshima et al., 2005*). Thus, we next examined whether Pan1p directly

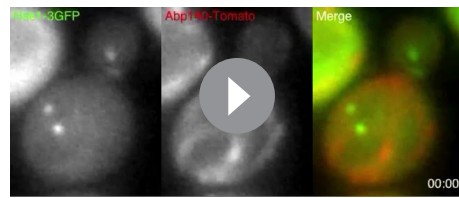

**Video 1.** Localization of Hse1-3GFP (left; green in merge) and Abp140-tdTomato (center; red in merge) in wild-type cells. The interval between frames is 1 s.

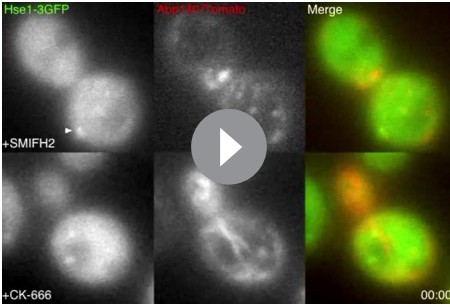

**Video 2.** Localization of Hse1-3GFP (left; green in merge) and Abp140-Tomato (center; red in merge) in wild-type cells treated with 25 $\mu$M SMIFH2 or 100 $\mu$M CK-666. Arrowheads indicate examples of Hse1p-labeled endosome. The interval between frames is 1 s.

mediates the interaction between vesicles and actin cables via its actin binding. To completely destroy the actin binding activity of Pan1p, we used a Pan1 C-terminal deletion mutant (*pan1△855–1480*), which lacks actin binding and Arp2/3-activating regions (*Duncan et al., 2001*; *Toshima et al., 2005*). Interestingly, combining the *pan1-18TA* and *pan1△855–1480* mutations (*pan1-18TA△855*) caused accumulation of Pan1-mCherry vesicles similar to *pan1-18TA* mutant, but formation of actin clumps were significantly suppressed (*Figure 5B,C*). The interaction between Pan1-mCherry-labeled vesicles and actin cables also decreased, but the vesicles still had an ability to bind actin cables (*Figure 4D*), suggesting the existence of additional actin-binding coat protein(s) that stabilize the association of vesicles with actin cables.

## Interaction between endocytic vesicles and early endosomes in the *pan1-18TA* phosphorylation site mutant

To be sure that this association is not merely a product of both being caught up in aberrant actin concentrations, we sought to isolate a mutant that could suppress actin clump formation in the *pan1-18TA* mutant. To this end, we deleted several actin-related genes involved in endocytic internalization in the *pan1-18TA* mutant and found three out of seven genes whose absence suppressed actin clump formation (*Figure 6A,B*). Deletion of Sla2p, a yeast Hip1R-related protein, in the *pan1-18TA* mutant decreased the fraction of cells containing actin clumps by 95% as detected by Abp1-mCherry in maximum-intensity projections of Z stacks (*Figure 6—figure supplement 1A,B*). These *pan1-18TA sla2△* double mutants exhibited the elongated actin tails originating from non-motile endocytic sites seen in *sla2△* (*Kaksonen et al., 2003*) (*Figure 6B*). Deletion of both of the yeast type I myosins, Myo3p and Myo5p, led to an 85% reduction and deletion of Sac6p, the yeast homologue of the actin filament bundling protein fimbrin, almost completely suppressed actin clump formation in the *pan1-18TA* mutant (*Figure 6—figure supplement 1A,B*). All three mutants showed a severe defect in the internalization of cortical actin patches (*Kaksonen et al., 2003*; *2005*), resulting in the loss of actin clumps in the *pan1-18TA* mutant. We then wished to examine the localization of early endosomes, using Hse1-GFP as a marker, in these double and triple mutants. Interestingly, Hse1p-labeled endosomes showed a change in localization to the cell periphery in all three suppressing mutants (*Figure 6B,C*). We also observed that Pan1-18TA patches accumulate near Hse1p-labeled endosomes at the cell periphery in *pan1-18TA sac6△* cells (*Figure 6D*). As reported previously (*Gheorghe et al., 2008*), deletion of the *SAC6* gene increased the lifetime of actin patches, but around half of the endocytic vesicles were able to internalize (*Figure 6—figure supplement 1C,D*), indicating that the formation of endocytic vesicles is delayed but eventually occurred. In contrast, in *pan1-18TA*

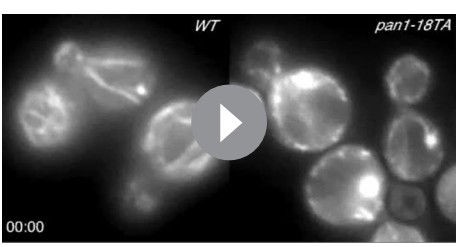

**Video 3.** Localization of Abp140-3GFP in wild-type and *pan1-18TA* cells. The interval between frames is 1 s.

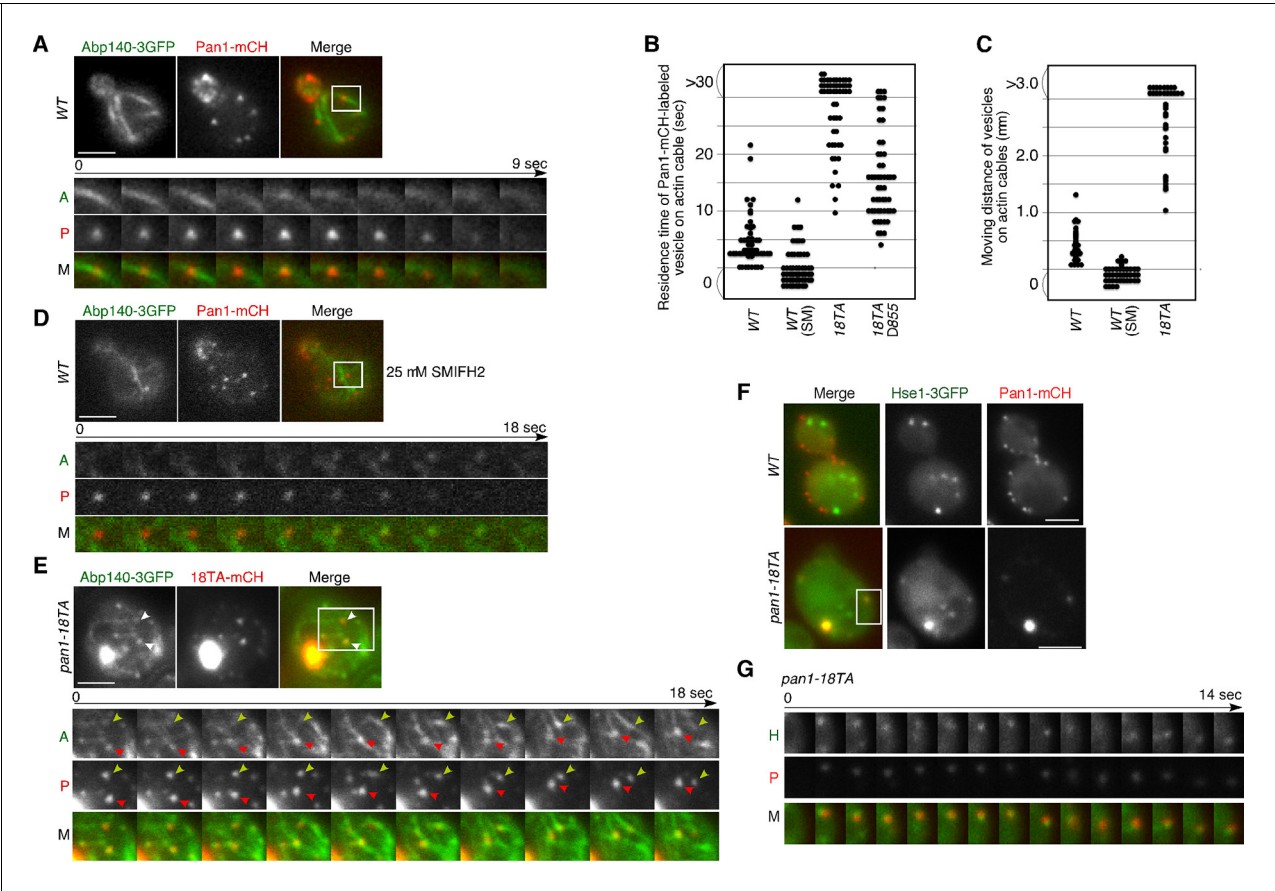

**Figure 4.** Interaction of Pan1p-containing compartments with actin cables and endosomes. (**A**) Localization of Abp140-3GFP and Pan1-mCherry in a wild-type cell. The lower panels correspond to a time series of a higher magnification view of the boxed area in the upper right image. (**B**) The residence time of Pan1-mCherry-labeled vesicles on an actin cable. The residence time was determined from 60 sequential two-dimensional images. $n = 52$ vesicles for each strain. Vesicles residing on cable over 30 s are indicated as >30 in the graph. (**C**) Moving distance of Pan1-mCherry-labeled vesicles on actin cables. To determine each moving distance, the distance that the center of the Pan1-mCherry fluorescence moves on an actin cable was calculated based on pixel coordinates (1 pixel = 64.5 nm). $n = 42$ vesicles for each strain. (**D**) Effect of the formin inhibitor SMIFH2 on the movement of Pan1-mCherry patches. Wild-type cells expressing Pan1-mCherry and Abp140-3GFP were grown to log phase at 25°C, treated with 25 μM SMIFH2 for 30 min at 25°C, and subsequently imaged at 1 s intervals. (**E**) The localization of Abp140-3GFP and Pan1-18TA-mCherry in a *pan1-18TA* cell. The lower panels are single focal plane images corresponding to a time series of a higher magnification view of the boxed area in the upper right image. Arrowheads indicate examples of Pan1p-containing compartments moving along an actin cable. Yellow or red arrowheads indicates different vesicles. Upper and middle panels show GFP and mCherry channels, respectively, and lower panel shows their merged images. (**F**) The localization of Hse1-3GFP and Pan1-mCherry in wild-type and *pan1-18TA* cells. (**G**) Time series of single patches in the boxed area in (**F**). Cells expressing Hse1-3GFP and Pan1-18TA-mCherry were grown to early to mid-logarithmic phase at 25°C in YPD medium and imaged at 1 s intervals. Scale bars, 2.5 μm.

The following figure supplement is available for figure 4:

**Figure supplement 1.** Actin cable dynamics in wild-type and *pan1-18TA* cells.

*sac6△* cells, the lifetime of actin patches were significantly increased and their internalizations were markedly decreased, but ~5.3% of patches were still internalized (*Figure 6—figure supplement 1D*). Thus, it seems that endocytic vesicle formation is not completely blocked in *pan1-18TA sac6△* cells. To examine if endocytic vesicles are formed in this double mutant, we explored the ultrastructure of these mutants using electron microscopy. Interestingly, we observed many vesicle-like structures (~40–60 nm) accumulating around the cell periphery in *pan1-18TA sac6△* cells, whereas in wild-type cells such structures were rarely detected (*Figure 6—figure supplement 2A–C*). We also observed that several endosome-like structures (~200–300 nm) associate with these vesicles in this mutant (*Figure 6—figure supplement 2B*). Quantification of these structures revealed that

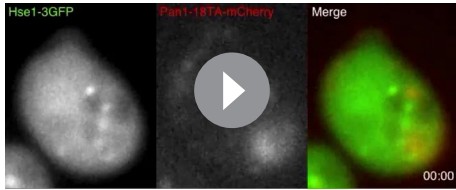

**Video 4.** Localization of Abp140-3GFP (left; green in merge) and Pan1-18TA-mCherry (center; red in merge) in *pan1-18TA* cells. The interval between frames is 1 s.

**Video 5.** Localization of Hse1-3GFP (left; green in merge) and Pan1-18TA-mCherry (center; red in merge) in *pan1-18TA* cells. Arrowheads indicate examples of colocalization. The interval between frames is 1 s.

endosome-like structures associate with ~1–3 small vesicles in two-dimensional cross sections (*Figure 6—figure supplement 2D*). Similarly to *pan1-18TA sac6△* cells, in *pan1-18TA sla2△* cells, Hse1p-labeled endosomes were observed at the cell periphery. *sla2△* cells are reported to have a severe defect in endocytic internalization, but the internalization is not completely inhibited (*Raths et al., 1993*) and endocytic coats are assembled on the plasma membrane (*Kaksonen et al., 2003*; *Skruzny et al., 2012*). Thus, in *pan1-18TA sla2△* cells, peripheral endosomes might interact with the occasionally formed endocytic vesicles at the cell periphery or with coat proteins accumulating on the plasma membrane. Examining Hse1-GFP in overlaid time-lapse images of *pan1-18TA sac6△* cells demonstrated its localization exclusively at the cell periphery and vacuolar membrane, and that peripheral endosomes are less motile (~17.8 nm/s), compared to prevacuolar ones (~189.0 nm/s) (*Figure 6E,F*). Since the *pan1-18TA sac6△* double mutant has apparently normal late endosomes at the vacuolar membrane, similar to wild-type cells (*Figure 6E*), the peripheral immotile endosomes observed in the mutant are likely early endosomes. Such peripheral endosomes were rarely observed in the *sac6△* mutant (*Figure 6E*), suggesting that they are due to the mutation of the phosphorylation sites in Pan1p and not just due to the defects in the actin cytoskeleton. LatA treatment had no effect on endosome localization in *pan1-18TA sac6△* cells (*Figure 6G*), supporting the contention that endocytic vesicles and early endosomes stably associate, independently of actin.

## Discussion

In yeast, endocytic vesicles require actin cables to mediate their transport to the early endosome. The vesicles move in a retrograde direction, from daughter toward mother cells (*Huckaba et al., 2004*; *Toshima et al., 2006*). This is distinct from most types of transport along actin cables, such as secretion and organellar division, that move from mother to daughter cells due to the action of myosin V motor proteins (Myo2p and Myo4p). Findings that endocytic vesicle movement occurs at the same velocity and direction as that of actin cables have suggested that endocytic vesicles remain fixed on the actin cables and move as a result of actin cable flow (*Girao et al., 2008*; *Huckaba et al., 2004*).

The molecular machinery that attaches endocytic vesicles to actin cables has not yet been elucidated, although a likely candidate would be an endocytic protein that binds F-actin. Several such proteins exist (*Engqvist-Goldstein and Drubin, 2003*). Considering that the association between endocytic vesicles and actin cables should be transient and controllable, Pan1p is an ideal candidate to mediate this interaction. Pan1p can bind directly to F-actin with high affinity (KD<0.5 μM) and its binding activity is regulated by phosphorylation through the Ark1/Prk1 kinases (*Toshima et al., 2005*). In this study, we demonstrated that a Pan1-18TA △855 mutant lacking its C-terminal actin binding and Arp2/3-activating regions partially suppressed the formation of the actin clump and reduced the interaction between Pan1p-residing vesicles and actin cables. Thus, Pan1p seems to be one of the key regulators that fixes vesicles to the actin cable and then dissociate from the cable and the vesicle upon phosphorylation (*Figure 7*). However, the ability of vesicles to bind to actin cables was not completely lost in the mutant, implying the existence of additional actin-binding coat protein(s) that stabilize vesicle association with actin cables. Sla2p, the yeast HIP1R, and Ent1p, the yeast epsin, bind to both the plasma membrane and F-actin via their N-terminal lipid-binding

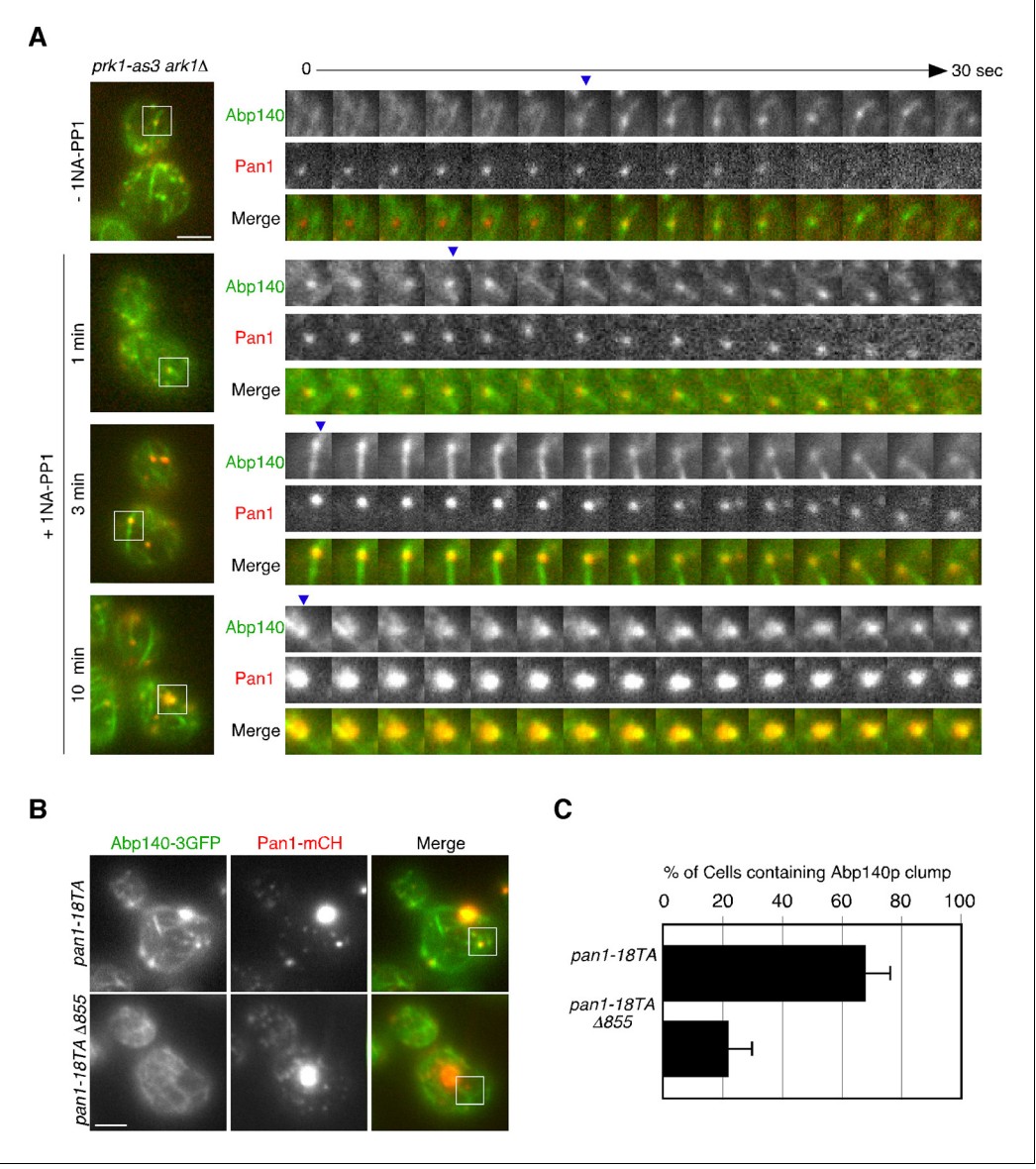

**Figure 5.** Interaction between endocytic vesicles and actin cables upon inhibition of Prk1p. (**A**) The left images represent single frames from movies of *ark1△ prk1-as3* mutant cells showing merged images of the GFP (Abp140p) and the mCherry (Pan1p) channel. The *ark1△ prk1-as3* mutant cells expressing Abp140-3GFP and Pan1-mCherry were grown to log phase at 25°C, treated with 100 μM 1NA-PP1 for the indicated time at 25°C, and subsequently imaged at 1 s intervals. A time series of single patches in the boxed area for each strain are shown in the right panels. Blue arrowheads indicate Pan1-mCherry-labeled vesicles associating with actin cables. Scale bar, 2.5 μm. (**B**) Localization of Abp140-3GFP and Pan1-mCherry in *pan1-18TA* and *pan1-18TA△855* cells. Scale bar, 2.5 μm. (**C**) Quantification of cells containing actin clumps. Cells expressing Abp140-3GFP were grown to log phase at 25°C and imaged. Data show mean ± SD from at least three experiments, with 50 cells counted for each strain per experiment.

domain and C-terminal actin-binding domain (*Skruzny et al., 2012*; *Sun et al., 2005*; *Yang et al., 1999*). A recent study showed that Sla2p and Ent1p interact redundantly with F-actin, and strains carrying a deletion of both proteins' actin-binding domains exhibit severe a defect in endocytosis (*Skruzny et al., 2012*). Although the defects caused by these mutants are predominantly observed in vesicle formation, these proteins could be responsible for the residual association of endocytic vesicles with actin cables in the Pan1-18TA actin-binding mutant.

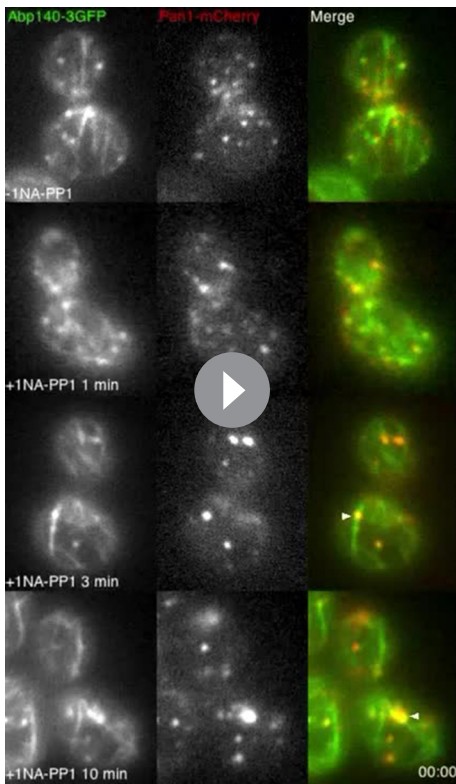

**Video 6.** Localization of Abp140-3GFP (left; green in merge) and Pan1-mCherry (center; red in merge) in *ark1△ prk1-as3* cells untreated or treated with 100 μM 1NA-PP1. Arrowheads indicate examples of vesicles associated with actin cables. The interval between frames is 1 s.

Early endosomes also associate with actin cables (*Chang et al., 2003*; *Toshima et al., 2006*), but the mechanism is still unknown. In the *pan1-18TA* mutant, endocytic vesicles stably associate with and move together with endosomes. This finding suggests that endocytic vesicles are capable of tethering to early endosomes, but are inefficient at fusing with them due to the inhibition of endocytic vesicle uncoating caused by Pan1p-dephosphorylation. Many tethering proteins localized at target organelles have been shown to directly interact with coat proteins of transport vesicles (*Cai et al., 2007*), supporting this idea. After phosphorylation of Pan1p by Ark1/Prk1 kinases, the endocytic vesicle is uncoated, making it possible to fuse to the early endosome. Thus the cycle of Pan1 phosphorylation could release endocytic vesicles from the actin cable precisely at the time of their fusion to the endosome, also allowing the endosome which is indirectly tethered to then move on and mature into a late endosome.

Our experiments may also permit a clearer ordering of the mechanistic steps of endocytosis. Many lines of evidence indicate that cargo transport from early to late endosomes is achieved by endosome maturation, which is a successive and rapid process accompanied by Rab5-Rab7 conversion (*Balderhaar and Ungermann, 2013*; *Nordmann et al., 2010*; *Poteryaev et al., 2010*; *Rink et al., 2005*). In yeast, attempts to visualize the conversion of Vps21p (yeast Rab5) to Ypt7p (yeast Rab7) on endosomes have so far been unsuccessful. Since Vps21p is localized not only at early, but also at late endosomes (*Toshima et al., 2014*), it is difficult to determine the point at which the early endosome ends and the late endosome begins. In the *pan1-18TA* mutant, localization of endosomal proteins was clearly divided into several groups, and endosomes at the early stage were more highly localized to actin clumps. This observation may be informative when considering the timing of several events comprising the endocytic pathway. For instance, subunits of the ESCRT-0, I, and II complex are highly localized but the subunit of ESCRT-III or Vps4p is localized only partially at actin clumps in the *pan1-18TA* mutant, suggesting that recruitment of early ESCRT components to the endosome occurs at a relatively early stage (*Figure 7*). Vps21p is a key regulator of early endocytic trafficking, being involved in fusion between early endosomes and the maturation of early to late endosomes (*Poteryaev et al., 2010*; *Rink et al., 2005*; *Russell et al., 2012*). Deletion of the *VPS21* gene results in accumulation of early endosomes in the cytosol (*Toshima et al., 2014*). Yet Vps21p exhibited only partial localization at actin clumps, suggesting that Vps21p is recruited to the endosomal membrane after the start of ESCRT complex formation. Vps26p, a component of the retromer complex, is rarely localized at actin clumps and is predominantly localized at the vacuolar membrane, suggesting that the retromer complex mediates retrograde transport to the Golgi at the late endosome stage (*Figure 7*). Although Rab5-Rab7 conversion is important for endosome maturation what triggers initiation of early-to-late endosome transition is still unknown. We demonstrated that early endosomes associate with actin filaments and then change their localization to the vacuolar membrane. This suggests that dissociation of early endosomes from the actin filaments might be a trigger for the initiation of endosome maturation.

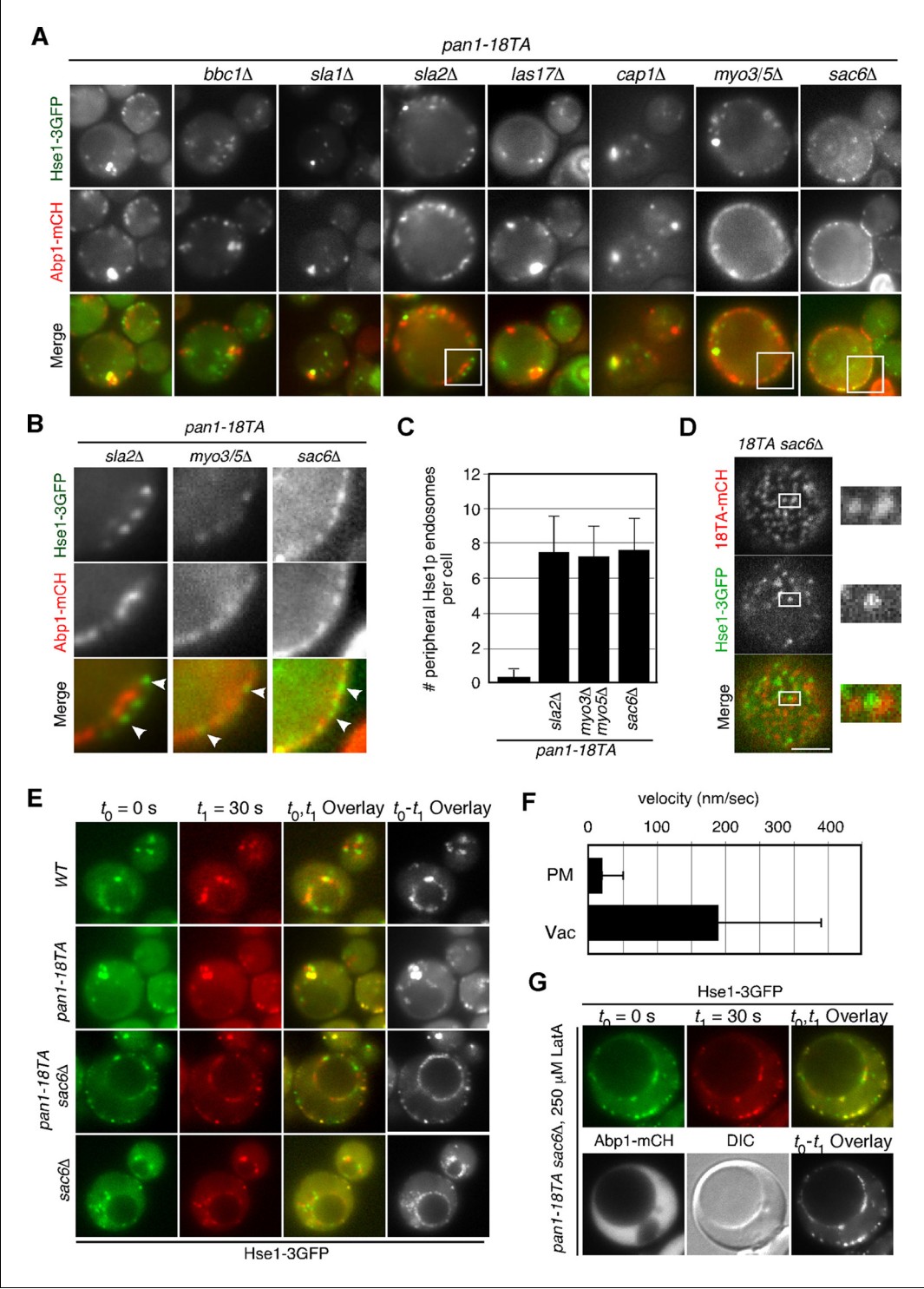

**Figure 6.** Interaction between endocytic vesicles and early endosomes in the *pan1-18TA* mutant. (**A**) Localization of Hse1-3GFP-labeled endosomes and actin structures in *pan1-18TA* double mutant cells. The *pan1-18TA* and double mutant cells expressing Hse1-3GFP and Abp1-mCherry were grown to early to mid-logarithmic phase at 25°C in YPD medium and observed by fluorescence microscopy. (**B**) Higher magnification view of the boxed areas in (**A**). Arrowheads indicate examples of Hse1p-containing endosomes at the plasma membrane. (**C**) Quantification of the number of Hse1p-containing endosomes localizing at the cell periphery in single focal plane images. Data show mean ± SD, with *n* = 50 cells counted for each strain. (**D**) Localization of Pan1-18TA-mCherry and Hse1-3GFP in *pan1-18TA sac6Δ* double mutant cells. (**E**) Movement of Hse1p-containing endosomes in *pan1-18TA, sac6△*,

*Figure 6 continued*

and the double mutant. Cells expressing Hse1-3GFP to visualize the endosomes were grown to log phase at 25°C, and imaged for 30 s at 1 s intervals. At time $t_0$ = 0 s, Hse1-GFP is shown in green, at $t_1$ = 30 s, Hse1-3GFP is shown in red. $t_0,t_1$ overlay shows overlay image of $t_0$ and $t_1$, and $t_0$-$t_1$ overlay shows overlay image of all 30 frames from $t_0$ (0 s) to $t_1$ (30 s). Scale bars, 2.5 μm. (**F**) Quantification of the velocity of Hse1p-containing endosomes at the cell periphery (PM) and the vacuolar membrane (Vac). (**G**) Localization of Hse1p-residing endosomes in the *pan1-18TA sac6△* mutant treated with 250 μM LatA. Cells treated with 250 μM LatA for 30 min at 25°C were imaged for 30 s at 1 s intervals. At time $t_0$ = 0 s, Hse1-GFP is shown in green, at $t_1$ = 30 s, Hse1-3GFP is shown in red. $t_0,t_1$ overlay shows overlay image of $t_0$ and $t_1$. Scale bars, 2.5 μm.

The following figure supplements are available for figure 6:

**Figure supplement 1.** Actin structures and actin patch dynamics in *pan1-18TA* double mutant cells.

**Figure supplement 2.** Ultrastructure of endocytic vesicles and endosomes observed in wild-type and *pan1-18TA sac6△* cells.

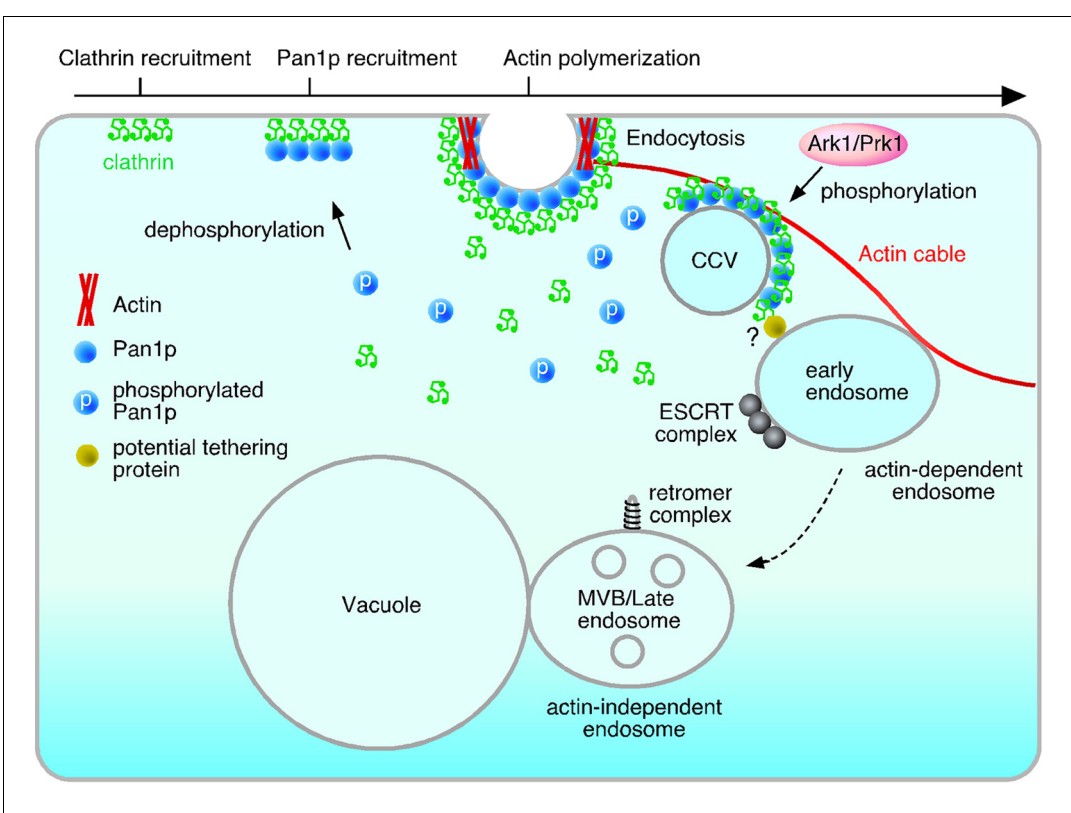

**Figure 7.** Model of the actin cable-mediated endocytic pathway. Unphosphorylated Pan1p on an endocytic vesicle binds to actin to fix the vesicle to the actin cable. After being pinched off from the membrane, the endocytic vesicle moves into the cytosol as a result of actin cable flow, and then, interacts with the early endosome via potential tethering protein. Pan1p phosphorylation by Ark1/Prk1 kinases causes dissociation of coat proteins and the actin cable from the endocytic vesicle, making it possible for the vesicle to fuse to the endosome. This also results in the dissociation of the actin cable and the early endosome, which then moves to the vacuolar membrane, and matures into a late endosome.

In mammalian cells, before endosomes move from the plasma membrane to the lysosome along microtubules, endosomes associate with the cortical actin cytoskeleton underlying the plasma membrane (*Aschenbrenner et al., 2004*; *Fernandez-Borja et al., 2005*). The interaction between endosomes and actin is regulated by RhoB GTPase, an upstream recruiter and activator of mammalian Dia1 and PRK1/PKN (*Fernandez-Borja et al., 2005*; *Mellor et al., 1998*). Expression of activated RhoB facilitates the association of early endosomes with cortical actin filaments, which prevents the transfer of endosomes to microtubules and inhibits further transport (*Fernandez-Borja et al., 2005*). The physiological importance of the interaction between endosomes and the actin cytoskeleton in mammalian cells has not been determined, but in analogy to our findings this might enable endosomes to fuse with endocytic vesicles more efficiently, at the same time preventing the progression of immature early endosomes to late endosomes.

In conclusion, our results suggest that phosphorylation of Pan1p regulates the interaction between endocytic compartments and the actin cytoskeleton. Clarifying the molecular mechanisms regulating the interaction between endocytic vesicles and endosomes, and endosomes and the actin cytoskeleton is important for elucidating the whole picture of transport from the formation of an endocytic vesicle to its fusion to an early endosome.

## Materials and methods

### Yeast strains, growth conditions, and plasmids

The yeast strains used in this study are listed in the strain list (*Supplementary file 1*). All strains were grown in standard rich medium (YPD) or synthetic medium (SM) supplemented with 2% glucose and appropriate amino acids. C-terminal GFP or mCherry tagging of proteins was performed as described previously (*Longtine et al., 1998*). The *pan1-18TA* mutant was integrated as follows: First, to create a *pan1* integration plasmid, the *Xmn*I-*Dra*I fragment of the *PAN1* gene was cloned into pBluescript II SK, and the *Sal*I fragment of the *LEU2* gene was inserted into the *Sal*I site 154-bp upstream of the *PAN1* ORF (*Toshima et al., 2005*). The mutated *Msc*I-*Nhe*I *pan1-18TA* fragments were used to replace the *PAN1* gene in the integration plasmid. To integrate *pan1* mutants at the endogenous locus, the integration plasmids were digested with *Sac*I and *Xba*I, and transformed into *pan1△::HIS3/PAN1* diploid strains. Integrated *pan1* mutants were selected on SC plates lacking leucine and sporulated to obtain *pan1-18TA* mutants. Phosphorylation site mutants were constructed by a PCR-based mutagenesis protocol (*Supplementary file 2*).

### Fluorescence microscopy

Fluorescence microscopy was performed using an Olympus IX81 microscope equipped with a x100/ NA 1.40 (Olympus) objective and Orca-AG cooled CCD camera (Hamamatsu, JAPAN), using Metamorph software (Universal Imaging). Simultaneous imaging of red and green fluorescence was performed using an Olympus IX81 microscope, described above, and an image splitter (Dual-View; Optical Insights) that divided the red and green components of the images with a 565-nm dichroic mirror and passed the red component through a 630/50 nm filter and the green component through a 530/30 nm filter. These split signals were taken simultaneously with one CCD camera, described above.

### Fluorescent labeling of $\alpha$-factor and endocytosis assays

Fluorescent labeling of α-factor was performed as described previously (*Toshima et al., 2006*). For endocytosis assays, cells were grown to an OD600 of ~0.5 in 0.5 ml YPD, briefly centrifuged, and resuspended in 20 μl SM with 5 μM Alexa Fluor-labeled α-factor. After incubation on ice for 2 hr, the cells were washed with ice-cold SM. Internalization was initiated by addition of SM containing 4% glucose and amino acids at 25°C.

### Western blot assays

Preparation of cell extracts was performed as described previously (*Toshima et al., 2005*). In brief, the cells grown in 200 ml YPD to OD600 of 1.0 were harvested by centrifugation, washed with dH$_2$O, resuspended to 1 ml of dH$_2$O, drop-frozen in liquid N$_2$, and powdered with mortar and pestle. The cell extracts were prepared using lysis buffer (50 mM Tri-HCl, pH 8.0, 150 mM NaCl, 8 M

Urea, 1% Triton X-100, phosphatase inhibitor cocktail). High molecular weight proteins over 100 K molecular weight were enriched using Amicon Ultra-0.5 100 K (Millipore), and phosphorylated proteins included in the fraction were enriched using Phos-tag Agarose (NARD Institute). The enrichment of phosphorylated proteins using Phos-tag Agarose was performed as previously (*Kinoshita-Kikuta et al., 2006*; *2009*). Immunoblot analysis was performed as described previously (*Toshima et al., 2005*). The chicken polyclonal antibody to GFP (GeneTex, GTX124117) was used at a dilution of 1:10000 and the HRP-conjugated rabbit polyclonal antibody to chicken IgY (Promega, G135A) at 1:10000 dilution was used as the secondary antibody. Immunoreactive proteins bands were visualized using the Western Lightning Plus ECL (PerkinElmer).

## $^{35}$S-labeled $\alpha$-factor internalization assay

Preparation and internalization of $^{35}$S-labeled $\alpha$-factor was performed as described previously (*Toshima et al., 2005*). Briefly, cells were grown to an OD600 of 0.3 in 50 ml YPD, briefly centrifuged and resuspended in 4 ml YPD containing 1% (w/v) BSA, 50 mM $KH_2PO_4$, pH 6.0, and 20 μg/ml uracil, adenine, and histidine. After adding $^{35}$S-labeled $\alpha$-factor, cell aliquots were withdrawn at various time points and subjected to a wash in pH 1 buffer to remove surface-bound $\alpha$-factor so internal $\alpha$-factor could be measured, or in pH 6 buffer to determine the total (internal and bound) $\alpha$-factor. The amount of cell-associated radioactivity after each wash was determined by scintillation counting. Each experiment was performed at least three times.

## Analysis of endosome motility

Endosome motility and velocity was analyzed using the ImageJ v1.32 software package. For quantification of endosome velocity, time-lapse images were acquired at 1 s intervals. To determine the velocity, the distance traveled by each endosome in 1 s was calculated based on pixel coordinates (1 pxl = 64 nm).

## Electron microscopy

Cells sandwiched between copper disks were frozen in liquid propane at -175°C and then freeze substituted with acetone containing 2% $OsO_4$ and 2% distilled water at -80°C for 48 hr. The samples were kept at -20°C for 4 hr and then at 4°C for 1 hr, and dehydrated in anhydrous acetone two times and 100% ethanol three times. After being infiltrated with propylene oxide (PO) two times the samples were put into a 70:30 mixture of PO and resin (Quetol-651) and then transferred to a fresh 100% resin, and polymerized at 60°C for 48 hr. The blocks were cut into 70-nm-thick sections, and the sections were mounted on copper grids. The specimens were stained with 2% uranyl acetate and Lead stain solution, and observed using a transmission electron microscope (JEM-1400Plus; JEOL).

## Acknowledgements

We thank A Masuda and N Yoshida for construction of plasmids and strains, and C Horikomi for analyzing data. This work was supported by JSPS KAKENHI GRANT #26440067 and a Takeda Science Foundation grant to JYT, as well as JSPS KAKENHI GRANT #25440054 and a Takeda Science Foundation grant to JT. Daria E Siekhaus is supported by EU grant PCIG12-GA-2012-334077.

## Additional information

### Funding

| Funder | Grant reference number | Author |
|---|---|---|
| Japan Society for the Promotion of Science | KAKENHI GRANT #26440067 | Junko Y Toshima |
| Japan Society for the Promotion of Science London | KAKENHI GRANT #25440054 | Jiro Toshima |
| European Union Grant | PCIG12-GA-2012-334077 | Daria Elisabeth Siekhaus |

The funders had no role in study design, data collection and interpretation, or the decision to submit the work for publication.

## Author contributions

JYT, JT, Conception and design, Acquisition of data, Analysis and interpretation of data, Drafting or revising the article; EF, MN, CK, YS, ME, Acquisition of data, Analysis and interpretation of data, Drafting or revising the article; DES, Conception and design, Analysis and interpretation of data, Drafting or revising the article

## Author ORCIDs

Jiro Toshima, http://orcid.org/0000-0003-3264-9843

## Additional files

**Supplementary files**
• Supplementary file 1. Yeast strains used in this study.

• Supplementary file 2. Primers used in this study.

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
