## [Decision Letter]

Thank you for submitting your work entitled "Pan1p regulates the interaction between endocytic vesicles, endosomes and the actin cytoskeleton" for peer review at *eLife*. Your submission has been favorably evaluated by Randy Schekman (Senior editor and Reviewing editor) and three reviewers.

The reviewers have discussed the reviews with one another and the Reviewing editor has drafted this decision to help you prepare a revised submission.

Summary of the work:

The manuscript addresses a fundamental question on endocytosis: How are movements and maturation of early endocytic compartments achieved and regulated? The results presented here provide important insights into the association of earliest endosomal compartments with the actin cytoskeleton, by studying how the coat protein Pan1 impacts interactions between endocytic vesicles, early endosomes and actin.

Essential revisions:

There is much to like about this manuscript. It is well written and addresses an important problem. The data are generally of high quality. There are issues that would need to be addressed, involving some additional experimentation and quantification, before the work would be appropriate for publication in *eLife*. How endocytic vesicles evolve into endosomal compartments is an important unanswered question in the field. The model developed by the authors is very interesting and if true, represents a significant advance for the field.

One concern is that the main experimental technique used in this work is 2D wide-field fluorescence microscopy, which may not be the best method to analyze endosomes, which move dynamically in 3 dimensions. Some of the key data (Figure 4) in this work lack quantification. In addition, some key results using mutants (Figure 5 and Figure 6) are a bit hard to interpret. Thus, the data are not yet adequate to support the conclusions.

Major concerns:

1) The claim that Pan1 is the key biological target for Ark1 and Prk1 kinases could use better support. The experiment done here, where 10 threonines within Sla1 are mutated to block its phosphorylation, should be backed up with some kind of experimental assessment that shows that these ten mutations completely block phosphorylation of Sla1. Without this kind of experimental evidence, it is impossible to tell whether the lack of endocytic phenotype in this mutant is due to Sla1 not being an important Ark1/Prk1 target, or if it is due to not all Ark1/Prk1 phosphorylation sites in Sla1 being mutated. Similarly, can the authors present evidence that the 18 threonines mutated in Pan1 are bona fide in vivo Ark1/Prk1 phosphorylation sites?

Nonetheless, given the similarity of the phenotype, it is likely that Ark1/Prk1 are the responsible kinases. However, that is not even relevant for the rest of their study, which deals with downstream effects of the phosphorylation. therefore, in the absence of the sort of results suggested above, the strong focus on Ark1/Prk1 in the interpretation of results as well as in Introduction and Discussion (last paragraph) should be reduced.

2) In Figure 3, the authors attempt to establish Hse1-3GFP as an early endosome marker in wild-type cells. However, Hse1 is also reported to be involved in late endosome to vacuole transport. Thus, some Hse1-3GFP puncta may not be early endosomes. In *pan1-18TA* mutant cells, endocytic internalization is largely defective, but it is not clear how the various endosomal compartments are affected. It does not seem reasonable to assume that Hse1-3GFP localizes only in early endosomes in this mutant. If Hse1-3GFP does not label only the early endosomes in *pan1-18TA* mutant, then the observation that "peripheral Pan1-18TA-mCherry patches colocalized and moved together with Hse1p-labeled endosomes," does not support the conclusion that "constitutive dephosphorylation of Pan1p, which inhibits endocytic vesicle uncoating, might lead to stable tethering between endocytic vesicles and early endosomes."

3) The last round of papers that looked at vesicle/endosome transport on actin cables was published before specific actin cytoskeleton drugs were developed, so the authors of those studies relied on LatA. Now, we have CK-666 (Arp2/3 complex inhibitor) and SMIFH2 (Formin FH2 Domain Inhibitor), so the model that vesicles move by attaching to cables and harnessing the power of actin flux could be tested more directly. Movement of these vesicles should be sensitive to SMIFH2 (or formin mutations) but not CK-666 if the model suggested in this paper is true.

4) The authors use the ESCRT0 component Hse1 as marker for early endosomes. They do not find the Rab5-orthologue Vps21, an established marker for early endosomes, to be present at significant amounts in *pan1-18TA*-induced actin clumps, and they further conclude form WT cells that Hse1 is present at very early endosomes, while Vps21 (and the CORVET complex) arrives later. This is in contrast to the general view that the Rab5/Vps21 marks early endosomes and is required for earliest fusion and maturation events (Zeigerer et al., Nature 2012, Cabrera et al., JBC 2013, Puchner et al., PNAS 2013 Arlt et al., MBoC 2014). The authors have previously published that initial steps in the yeast endocytic network are not dependent on Rab5, however, they have also seen that about 80% of Vps21 endosomes colocalize with Hse1 (Toshima et al., Nature Communications 2014). It is therefore surprising that Vps21 would not be able to mark the here studied earliest endosomes. Given its importance, I suggest substantiating the role of Hse1 as an early marker, provide a more precise definition of the early endosomes studied here, and discuss the results with respect to the literature on Vps21. The exclusion of Vps21 from early steps seems to be based on the quantification of actin clumps that contain the GFP-tagged protein of interest (Figure 2). However, this does not take into account differences in the amount and lifetime of the individual proteins. I would rather suggest quantifying and comparing the percentage of GFP-fused protein that is colocalizing with actin clumps. To clarify the identity of the early endosomes that associate with actin, I suggest the measurements of endosome velocity marked by Hse1-GFP/Vps26-mCherry, with and without LatA, (subsection “Actin-dependent motility of Hse1p-residing endosomes”) to be also done with GFP-Vps21/Vps26-mCherry.

5) Data shown in Figure 4 should be quantified. One of the main claims of the paper is that the *pan1-18TA* mutation causes stable association of peripheral actin patches with cables. This should be quantified meticulously, and numbers for run length and amount of time spent associated with cables should be reported for patches in wild-type cells and *pan1-18TA* cells. Care should also be taken to define the percent of endocytic vesicles that associate with actin cables in the wild-type and mutant conditions.

6) The data shown in Figure 5 and Figure 6 with *pan1-18TA sac6∆* are hard to interpret. Deletion of SAC6 alone is known to cause substantial defects in endocytic vesicle formation and scission. Thus, it is not clear if and how endocytic vesicles are formed in *pan1-18TA sac6∆* cells. For Figure 6, the authors should provide evidence that the small vesicles are endocytic vesicles and the bigger vesicles are early endosomes. In addition, a larger sample size should be analyzed for Figure 6, with quantification.

7) It seems that a more straightforward method to test the authors' model shown in Figure 7 than using *pan1-18TA sac6* would be a chemical genetics approach (Bishop et al., 2001), which rapidly modulates Ark1/Prk1p kinase activity in vivo (Sekiya-Kawasaki et al.). If the authors' model is correct, upon a rapid inhibition of Ark1/Prk1 kinase activity by 1NA-PP1, Pan1p should no longer be phosphorylated and the endocytic vesicles should be fixed to the actin cables near the cell periphery. However, the data shown in Sekiya-Kawasaki et al. did not seem to suggest that this is the case.

8) The scope of the final Results section (”Interaction between endocytic vesicles and early endosomes in the *pan1-18TA* phosphorylation site mutant”) is a bit unclear. The authors search for mutants that suppress clump formation in *pan1-18TA*, to distinguish tethering of vesicles to endosomes from association of both with actin, if I understand correctly. They find three such mutants that all show peripheral, non-motile endosomes. Firstly, I miss clear evidence that these endosomes are less motile than prevacuolar ones. Second, the authors conclude from their data that these early endosomes associate with vesicles independently of actin. The problem is that *sac6∆*, as well as the other two suppressor deletions, are known to inhibit endocytic internalization (as the authors correctly cite) – why would there be any endocytic vesicles present? Could there be another interpretation for the presence of peripheral endosomes in these mutants? Could the vesicles seen in EM (Figure 6) be other than primary endocytic vesicles?

9) The authors interpret their results to support the model that Pan1 mediates interaction between vesicles and actin cables (for example, Discussion, start of second paragraph). However, the experiments in Figure 4 do not distinguish between a direct role for Pan1 as the bridging molecule or an indirect role through effects on uncoating that allow for other actin-binding coat proteins to stabilize vesicle association with actin cables. One approach to address this issue might be to introduce actin-binding mutations into Pan1-18TA, such as the WH2-like domain KE mutations described in Toshima et al., 2005. Such mutations are expected to disrupt actin cable binding but not vesicle uncoating. Although these mutations suppress the actin aggregation phenotype, it should still be possible to assess vesicle interaction with actin cables. Alternatively, the actin-binding mutations could be introduced into an otherwise wild-type Pan1 and tested for effects on vesicle association with cables.

[Editors' note: further revisions were requested prior to acceptance, as described below.]

Thank you for submitting your work entitled "Pan1p regulates regulates the interaction between endocytic vesicles, endosomes and the actin cytoskeleton" for consideration by *eLife*. Your article has been reviewed by three peer reviewers, and the evaluation has been overseen by Randy Schekman as the Senior Editor and Reviewing Editor. One of the three reviewers has agreed to share his identity: Greg Payne.

The reviewers have discussed the reviews with one another and the Reviewing Editor has drafted this decision to help you prepare a revised submission.

Summary:

This manuscript describes how primary endocytic vesicles associate with and are transported along actin cables. The transiency of this interaction appears to be regulated by phosphorylation of Pan1. Furthermore, the authors find that early endosomes also associate with actin cables. They also present evidence that tethering interactions between early endosomes and primary vesicles takes place during this phase of actin association. The authors use yeast cells expressing a Pan1-phosphorylation mutant, which retards these transient events, to dissect the actin-cable association of endosomes and vesicles. They back up their interpretations of phenotypes by experiments in wild type cells including transient drug treatments.

The authors have addressed the reviewers' major concerns in the revised manuscript. What has particularly improved the manuscript is that they clarified the role of Hse1 as an endosomal marker (and, thus, the identity of the endosomes they study), they quantified the association of vesicles and endosomes with actin, and finally, the experiments with analogue-sensitive Prk1 substantiated the phosphorylation-dependent dissociation of vesicles from cables.

We do think the manuscript is a significant contribution to our understanding of the early events in the endosomal pathway. However, several concerns remain that could be addressed by changes in the text

Essential revisions:

1) The conclusion that constitutive dephosphorylation, which inhibits uncoating, leads to stable tethering of primary vesicles to endosomes (end of subsection “Pan1p-labeled endocytic vesicles associate with actin cables in the *pan1-18TA* mutant”) is based on the observation that in the *pan1-18TA* mutant, endocytic vesicles and endosomes move together along cables. While overall, their results do suggest that endocytic vesicles and endosomes, before fusion, interact and tether while both associated to actin cables, the conclusion that dephosphorylation leads to their stable tethering, seems a bit over interpreted. In this mutant, both vesicles and endosomes are associated with actin for much longer, therefore their long-time tethering may be due to crowding or other effects, rather than the fact that dephosphorylation prevents uncoating of the vesicles.

The evidence supporting an association of endosomes with endocytic vesicles in the absence of actin is also weak. It seems clear that there is an increase in peripheral endosomes in cells lacking actin clumps (Figure 6), or treated with LatA (6G), but that this localization is due to tethering with vesicles in not established. The data in 6D are not very convincing – a single static image of double mutant cells with no quantification and no comparison to single mutants. It may be a challenge for the authors to provide additional support for stable association in live cells since tracking association during movement may be difficult because of the limited motility of the peripheral endosomes. In the absence of additional data, the authors could rewrite so that association via tethering is a proposal rather than an interpretation/conclusion and either take out the tether in Figure 7 or put a question mark next to it.

Furthermore, the use of the *sac6delta/pan1-18TA* mutant to show that tethering of early endosomes with vesicles is not just due to actin clumping but independent of actin, is not yet entirely clear-. If I understand correctly, this finding is based on the fact that in the double mutant, early endosomes accumulate in the cell periphery, and that must be due to their association with vesicles. However, they also see endosomes in the cell periphery of *sla2∆* cells, which is known to totally block endocytic vesicle formation (Kaksonen 2003, Skruzny 2012). How are these endosomes held at the cell periphery? (subsection “Interaction between endocytic vesicles and early endosomes in the *pan1-18TA* phosphorylation site mutant”).

2) The authors failed to address the major concern that they are using 2D wide field microscopy to study endosomes, which move in three dimensions. Use of three-dimensional microscopy could improve the quality of the data presented in Figure 4 in particular. While I am disappointed that the authors still present only 2D microscopy, I believe that the data presented in Figure 4 are sufficient to support the authors' conclusions. A comment on this limitation is requested.

3) The layout of Figure 1, Figure 4 and Figure 6 should be changed such that the panels appear in alphabetical order. For example, panel F in Figure 6 seems to be out of place, coming after B and C but before D on the page.

4) In the text (subsection “Interaction between endocytic vesicles and early endosomes in the *pan1-18TA* phosphorylation site mutant”), the authors claim that deletion of SLA2 leads to 95% reduction in actin clumps, but the representative image of a *sla2Δ* cell in Figure 6 clearly has an Abp1-labeled actin clump in the cytoplasm. In contrast, the *myo3Δ/myo5Δ* cell shown has no noticeable actin clump, even though the authors say in the text that clumps form more readily in this mutant. Two comments on this confusion: first, the authors should explicitly mention how they are quantifying internal clump formation; are the clumps 95% less bright in *sla2Δ* cells? Or do they form 95% less frequently? Second, perhaps a better representative image could be chosen, such that the numbers stated in the text are borne out in the data shown in the figure.

5) In the Discussion, first paragraph, the authors cite the vesicle formation defects in *sla2* and *ent1* actin-binding mutants as evidence that these proteins function before transport of endocytic vesicles on actin cables. However, predominant effects of the mutants on vesicle formation could obscure latter roles in vesicle transport and thus do not preclude such roles. Consequently, Sla2 and Ent1 could be responsible for the residual association of endocytic vesicles with actin cables in the Pan1-18TA actin-binding mutant. It might be worth editing the text in the Discussion to allow for this possibility.

---

## [Author Response]

*Essential revisions: There is much to like about this manuscript. It is well written and addresses an important problem. The data are generally of high quality. There are issues that would need to be addressed, involving some additional experimentation and quantification, before the work would be appropriate for publication in eLife. How endocytic vesicles evolve into endosomal compartments is an important unanswered question in the field. The model developed by the authors is very interesting and if true, represents a significant advance for the field. One concern is that the main experimental technique used in this work is 2D wide-field fluorescence microscopy, which may not be the best method to analyze endosomes, which move dynamically in 3 dimensions. Some of the key data (Figure 4) in this work lack quantification. In addition, some key results using mutants (Figure 5 and Figure 6) are a bit hard to interpret. Thus, the data are not yet adequate to support the conclusions. 1) The claim that Pan1 is the key biological target for Ark1 and Prk1 kinases could use better support. The experiment done here, where 10 threonines within Sla1 are mutated to block its phosphorylation, should be backed up with some kind of experimental assessment that shows that these ten mutations completely block phosphorylation of Sla1. Without this kind of experimental evidence, it is impossible to tell whether the lack of endocytic phenotype in this mutant is due to Sla1 not being an important Ark1/Prk1 target, or if it is due to not all Ark1/Prk1 phosphorylation sites in Sla1 being mutated. Similarly, can the authors present evidence that the 18 threonines mutated in Pan1 are bona fide* in vivo

*Ark1/Prk1 phosphorylation sites?*

In accordance with the reviewers’ suggestion, using Phos-tag agarose, we purified phosphorylated Sla1p from wild-type and *sla1-10TA* cells and examined the in vivo phosphorylation state of Sla1-10TA. In Figure 1—figure supplement 1 in the new manuscript, we show that wild-type Sla1p is phosphorylated in vivo, whereas phosphorylation of Sla1-10TA mutant is completely inhibited. We also show that Pan1p phosphorylation is also inhibited in *pan1-18TA* mutant (Figure 1).

*Nonetheless, given the similarity of the phenotype, it is likely that Ark1/Prk1 are the responsible kinases. However, that is not even relevant for the rest of their study, which deals with downstream effects of the phosphorylation. therefore, in the absence of the sort of results suggested above, the strong focus on Ark1/Prk1 in the interpretation of results as well as in Introduction and Discussion (last paragraph) should be reduced.*

In accordance with the reviewers’ excellent suggestion, we performed experiment using analogue-sensitive mutant of Prk1p in cells lacking Ark1p (*ark1*△ *prk1-as3*) (please see reviewers’ comment #7). This experiment showed that transient inhibition of Prk1p’s kinase activity caused stable association of endocytic vesicles to actin cables that resembled the phenotype observed in the *pan1-18TA* mutant (Figure 5). This result supports the idea that phosphorylation by Ark1/Prk1 kinases regulates interaction between vesicles and actin cables. However, since we agree the reviewers’ opinion that we emphasize Ark1/Prk1 too much without enough evidences, we modified the sentences in Introduction and Discussion as follows;

In the Introduction, the sentence “Thus, Ark1/Prk1 kinases seem to phosphorylate Pan1p to regulate the interaction between endocytic compartments and the actin cytoskeleton.” was changed to “phosphorylation of Pan1p seems to regulate the interaction between ~”.

In the Discussion, the sentence “the Ark1/Prk1 kinases phosphorylate Pan1p to regulate the interaction ~” was changed to “phosphorylation of Pan1p regulates ~”.

*2) In Figure 3, the authors attempt to establish Hse1-3GFP as an early endosome marker in wild-type cells. However, Hse1 is also reported to be involved in late endosome to vacuole transport. Thus, some Hse1-3GFP puncta may not be early endosomes. In pan1-18TA mutant cells, endocytic internalization is largely defective, but it is not clear how the various endosomal compartments are affected. It does not seem reasonable to assume that Hse1-3GFP localizes only in early endosomes in this mutant. If Hse1-3GFP does not label only the early endosomes in pan1-18TA mutant, then the observation that "peripheral Pan1-18TA-mCherry patches colocalized and moved together with Hse1p-labeled endosomes," does not support the conclusion that "constitutive dephosphorylation of Pan1p, which inhibits endocytic vesicle uncoating, might lead to stable tethering between endocytic vesicles and early endosomes."*

We agree with the reviewer that some Hse1-3GFP puncta may not be early endosomes. As we showed in Figure 3—figure supplement 1, Hse1p is widely localized from early to late endosomes and partially colocalized with Vps26p at late endosomes. We also showed that similar levels of Hse1p colocalized with Vps26p in *pan1-18TA* mutant (Figure 3—figure supplement 1). Thus, we changed the sentence “constitutive dephosphorylation of Pan1p, which inhibits endocytic vesicle uncoating, might lead to stable tethering between endocytic vesicles and early endosomes.” to “~ stable tethering between endocytic vesicles and endosomes.”

*3) The last round of papers that looked at vesicle/endosome transport on actin cables was published before specific actin cytoskeleton drugs were developed, so the authors of those studies relied on LatA. Now, we have CK-666 (Arp2/3 complex inhibitor) and SMIFH2 (Formin FH2 Domain Inhibitor), so the model that vesicles move by attaching to cables and harnessing the power of actin flux could be tested more directly. Movement of these vesicles should be sensitive to SMIFH2 (or formin mutations) but not CK-666 if the model suggested in this paper is true.*

We examined the effects of CK-666 and SMIFH2 on the movement of endocytic vesicles and endosomes. As shown in Figure 3 and Figure 4, we found that the movement of endocytic vesicles and endosomes are sensitive to SMIFH2 but not CK-666, as the reviewer suggested.

*4) The authors use the ESCRT0 component Hse1 as marker for early endosomes. They do not find the Rab5-orthologue Vps21, an established marker for early endosomes, to be present at significant amounts in pan1-18TA-induced actin clumps, and they further conclude form WT cells that Hse1 is present at very early endosomes, while Vps21 (and the CORVET complex) arrives later. This is in contrast to the general view that the Rab5/Vps21 marks early endosomes and is required for earliest fusion and maturation events (Zeigerer et al., Nature 2012, Cabrera et al., JBC 2013, Puchner et al., PNAS 2013 Arlt et al., MBoC 2014). The authors have previously published that initial steps in the yeast endocytic network are not dependent on Rab5, however, they have also seen that about 80% of Vps21 endosomes colocalize with Hse1 (Toshima et al., Nature Communications 2014). It is therefore surprising that Vps21 would not be able to mark the here studied earliest endosomes. Given its importance, I suggest substantiating the role of Hse1 as an early marker, provide a more precise definition of the early endosomes studied here, and discuss the results with respect to the literature on Vps21. The exclusion of Vps21 from early steps seems to be based on the quantification of actin clumps that contain the GFP-tagged protein of interest (Figure 2). However, this does not take into account differences in the amount and lifetime of the individual proteins. I would rather suggest quantifying and comparing the percentage of GFP-fused protein that is colocalizing with actin clumps. To clarify the identity of the early endosomes that associate with actin, I suggest the measurements of endosome velocity marked by Hse1-GFP/Vps26-mCherry, with and without LatA, (subsection “Actin-dependent motility of Hse1p-residing endosomes”) to be also done with GFP-Vps21/Vps26-mCherry.*

To determine the relative localization of Hse1p and Vps21p at early stages of endocytosis, we slowed endocytic traffic by removing glucose from the culture medium (Aoh et al., MBoC, 2011), and compared their localization with internalized A594-α-factor. Interestingly, we found that Hse1p colocalizes with A594-α-factor at slightly higher amounts than Vps21p at 10 min after α-factor internalization (Figure 3—figure supplement 2). This result is consistent with a recent study by Arlt et al. (MBoC, 2015). In the paper, they carefully compared the recruitment timing of several endosomal proteins, including ESCRT subunits and Vps21p and reported that an ESCRT-I subunit, Vps23, is recruited to endosomes earlier than Vps21p. We also quantified the percentage of GFP-fused Vps21p that colocalizes with actin clumps, as the reviewer suggested, and found only ~8% of GFP-Vps21p is localized to actin clumps whereas ~22% of Hse1-GFP is localized to actin clumps (data not shown in the manuscript). Thus, it seems to be reasonable that Hse1p, another ESRCT-I subunit, is recruited to early endosomes before Vps21p, and that Hse1p localizes to actin clumps more than Vps21p.

Additionally, in accordance with the reviewers’ suggestion, we examined the velocity of endosomes marked by GFP-Vps21 with or without Vps26-mCherry. Similar to Hse1p-residing endosomes, the velocity of GFP-Vps21 endosomes not labeled with Vps26-mCherry was decreased by LatA treatment, whereas those labeled with Vps26-mCherry was not significantly affected (Figure 3—figure supplement 2). Taken with the previous results (colocalization with Vps21p and recruitment of ESCRT-1 complex to early endosomes), we concluded that Hse1p is suitable as an early endosome marker, as well as Vps21p.

*5) Data shown in Figure 4 should be quantified. One of the main claims of the paper is that the pan1-18TA mutation causes stable association of peripheral actin patches with cables. This should be quantified meticulously, and numbers for run length and amount of time spent associated with cables should be reported for patches in wild-type cells and pan1-18TA cells. Care should also be taken to define the percent of endocytic vesicles that associate with actin cables in the wild-type and mutant conditions.*

According to the reviewers’ suggestion, we added quantified data in Figure 4. With regard to percent of endocytic vesicles that associate with actin cables, we recently reported that over 80% of actin patches were internalized actin cables at the internalization step of endocytosis (Toshima et al., JCS, 2015). We also examined their association in *pan1-18TA* mutant, and found that over 80% of Pan1-mCherry-labeled endocytic vesicles also associated with and internalized along actin cables in *pan1-18TA* mutant. To explain this result, we have added some sentences to the text (subsection “Pan1p-labeled endocytic vesicles associate with actin cables in the *pan1-18TA* mutant”).

*6) The data shown in Figure 5 and Figure 6 with pan1-18TA sac6∆ are hard to interpret. Deletion of SAC6 alone is known to cause substantial defects in endocytic vesicle formation and scission. Thus, it is not clear if and how endocytic vesicles are formed in pan1-18TA sac6∆ cells. For Figure 6, the authors should provide evidence that the small vesicles are endocytic vesicles and the bigger vesicles are early endosomes. In addition, a larger sample size should be analyzed for Figure 6, with quantification.*

To examine the defects of endocytic vesicle formation and scission quantitatively in *sac6*△ and *pan1-18TA sac6∆* mutant, we quantified actin patch lifetime and percentage of vesicle internalization. As reported previously (Gheorghe et al., 2008), deletion of the *SAC6* gene increased actin patch lifetime, but around half of endocytic vesicles were able to move inside the cells, indicating that though the formation of endocytic vesicles is delayed it eventually occurred (Figure 6—figure supplement 1). In contrast, in *pan1-18TA sac6*△ cells, the lifetime of actin patches were significantly increased and their internalizations were rarely observed. Yet ~5.3% of patches were still internalized (Figure 6—figure supplement 1) supporting the conclusion that endocytic vesicle formation is not completely blocked in *pan1-18TA sac6∆* cells.

We also performed immuno-EM to identify the small vesicles and the bigger vesicles observed in the EM images as endocytic vesicles and early endosomes, but we could not obtain a clear result because our antibodies did not work on the EM samples, probably due to the low expression of Pan1p or Hse1p on the vesicles. Thus, as reviewer #2 suggested in comment #5, we moved the Figure to the supplementary file (Figure 6—figure supplement 2) and de-emphasized the result in the new manuscript (subsection “Interaction between endocytic vesicles and early endosomes in the *pan1-18TA* phosphorylation site mutant”). In addition, we have added quantified data in Figure 6—figure supplement 2.

7) It seems that a more straightforward method to test the authors' model shown in Figure 7 than using pan1-18TA sac6 would be a chemical genetics approach (Bishop et al., 2001), which rapidly modulates Ark1/Prk1p kinase activity in vivo (Sekiya-Kawasaki et al.). If the authors' model is correct, upon a rapid inhibition of Ark1/Prk1 kinase activity by 1NA-PP1, Pan1p should no longer be phosphorylated and the endocytic vesicles should be fixed to the actin cables near the cell periphery. However, the data shown in Sekiya-Kawasaki et al.

*did not seem to suggest that this is the case.*

In accordance with the reviewers’ suggestion, we performed an experiment using an analogue-sensitive mutant of Prk1p in cells lacking Ark1p (*ark1*△ *prk1-as3*). This mutant enabled us to investigate the direct and immediate consequence of Prk1p inactivation for the interaction between endocytic vesicles and actin cables. As shown in Figure 5, we found that, at 1 min after treatment of the mutant with 100 μM 1NA-PP1, endocytic vesicles stably associated with, and moved on actin cables. Then, small aggregates containing Pan1p that associated with actin cables were formed by 3 min, and a large actin clump that stably associates with actin cables was formed in the mutant after 10 min (Figure 5). These observations support the idea that Pan1p phosphorylation by Ark1/Prk1 kinases promotes dissociation of endocytic vesicles from actin cables.

*8) The scope of the final Results section (”Interaction between endocytic vesicles and early endosomes in the pan1-18TA phosphorylation site mutant”) is a bit unclear. The authors search for mutants that suppress clump formation in pan1-18TA, to distinguish tethering of vesicles to endosomes from association of both with actin, if I understand correctly. They find three such mutants that all show peripheral, non-motile endosomes. Firstly, I miss clear evidence that these endosomes are less motile than prevacuolar ones.*

We have added quantified data for the endosome velocity in Figure 6.

*Second, the authors conclude from their data that these early endosomes associate with vesicles independently of actin. The problem is that sac6∆, as well as the other two suppressor deletions, are known to inhibit endocytic internalization (as the authors correctly cite) – why would there be any endocytic vesicles present? Could there be another interpretation for the presence of peripheral endosomes in these mutants? Could the vesicles seen in EM (Figure 6) be other than primary endocytic vesicles?*

To examine of if endocytic vesicles are formed in *pan1-18TA sac6*∆ cells, we examined the lifetime and dynamics of actin patches in *sac6*∆ and *pan1-18TA sac6*∆ cells (please see the response to comment #6). As shown in Figure 6—figure supplement 1, the lifetime of actin patches were significantly increased and their internalizations were rarely observed, but ~5.3% of patches were still internalized. Thus, it seems that endocytic vesicle formation is not completely blocked in *pan1-18TA sac6∆* cells. With regard to the EM data, we have moved the Figure to the supplementary file and de-emphasized the result, because we could not confirm the exact identity of the vesicle-like and endosome-like structures through antibody stains.

*9) The authors interpret their results to support the model that Pan1 mediates interaction between vesicles and actin cables (for example, Discussion, start of second paragraph). However, the experiments in Figure 4 do not distinguish between a direct role for Pan1 as the bridging molecule or an indirect role through effects on uncoating that allow for other actin-binding coat proteins to stabilize vesicle association with actin cables. One approach to address this issue might be to introduce actin-binding mutations into Pan1-18TA, such as the WH2-like domain KE mutations described in Toshima et al., 2005. Such mutations are expected to disrupt actin cable binding but not vesicle uncoating. Although these mutations suppress the actin aggregation phenotype, it should still be possible to assess vesicle interaction with actin cables. Alternatively, the actin-binding mutations could be introduced into an otherwise wild-type Pan1 and tested for effects on vesicle association with cables.*

To determine whether Pan1p directly mediates the interaction between endocytic vesicles and actin cables, we prepared the Pan1-18TA mutant lacking its C terminal actin binding and Arp2/3-activating regions (Duncan et al., NCB, 2001; Toshima et al., NCB, 2005). Interestingly, as the reviewer expected, *pan1-18TA*△*855* caused accumulation of vesicles containing Pan1p similar to that seen in the *pan1-18TA* mutant, but the formation of actin clumps was significantly suppressed (Figure 5). The interaction between vesicles and actin cables also decreased, but the ability of endocytic vesicles to bind to actin cables was not completely lost (Figure 4). Thus, it seems that other actin-binding coat protein(s) that stabilizes vesicle association with actin cables exist, in addition to Pan1p.

[Editors' note: further revisions were requested prior to acceptance, as described below.]

*Essential revisions: 1) The conclusion that constitutive dephosphorylation, which inhibits uncoating, leads to stable tethering of primary vesicles to endosomes (end of subsection “Pan1p-labeled endocytic vesicles associate with actin cables in the pan1-18TA mutant”) is based on the observation that in the pan1-18TA mutant, endocytic vesicles and endosomes move together along cables. While overall, their results do suggest that endocytic vesicles and endosomes, before fusion, interact and tether while both associated to actin cables, the conclusion that dephosphorylation leads to their stable tethering, seems a bit over interpreted. In this mutant, both vesicles and endosomes are associated with actin for much longer, therefore their long-time tethering may be due to crowding or other effects, rather than the fact that dephosphorylation prevents uncoating of the vesicles. The evidence supporting an association of endosomes with endocytic vesicles in the absence of actin is also weak. It seems clear that there is an increase in peripheral endosomes in cells lacking actin clumps (Figure 6), or treated with LatA (6G), but that this localization is due to tethering with vesicles in not established. The data in 6D are not very convincing* –

*a single static image of double mutant cells with no quantification and no comparison to single mutants. It may be a challenge for the authors to provide additional support for stable association in live cells since tracking association during movement may be difficult because of the limited motility of the peripheral endosomes. In the absence of additional data, the authors could rewrite so that association via tethering is a proposal rather than an interpretation/conclusion and either take out the tether in Figure 7 or put a question mark next to it.*

In accordance with the reviewers’ suggestion, we rewrote the sentence which said “constitutive dephosphorylation, which inhibits uncoating, leads to stable tethering of primary vesicles to endosomes.” to “in the *pan1-18TA* mutant, endocytic vesicles and endosomes interact before fusion while they are both associated with actin cables, potentially tethering them together.” We also modified Figure 7 as reviewer suggested, indicating a protein as a “potential tethering protein” in the legend and placing a question mark next to it in the figure.

*Furthermore, the use of the sac6∆ /pan1-18TA mutant to show that tethering of early endosomes with vesicles is not just due to actin clumping but independent of actin, is not yet entirely clear-. If I understand correctly, this finding is based on the fact that in the double mutant, early endosomes accumulate in the cell periphery, and that must be due to their association with vesicles. However, they also see endosomes in the cell periphery of sla2∆ cells, which is known to totally block endocytic vesicle formation (Kaksonen 2003, Skruzny 2012). How are these endosomes held at the cell periphery? (subsection “Interaction between endocytic vesicles and early endosomes in the pan1-18TA phosphorylation site mutant”).*

As the reviewers’ suggested, *sla2*△ cells are reported to have a severe defect in endocytic internalization, but the internalization is not completely blocked at 24^o^C as shown in Figure 1 (Raths et al., JCB, 1993). Thus, similarly to *pan1-18TA sac6*△ cells, endocytic vesicle formation might not be completely blocked in *pan1-18TA sla2∆* cells and peripheral endosomes could interact with those endocytic vesicles. As another possibility, in *sla2*△ cells, although endocytic internalization is severely inhibited, endocytic coats are assembled on the plasma membrane (Kaksonen et al., Cell, 2003; Skruzny et al., PNAS, 2012). Thus, peripheral endosomes might interact with coat proteins accumulating on the plasma membrane in *pan1-18TA sla2*△ cells. To explain about this, we have added some sentences in the Results section (subsection “Interaction between endocytic vesicles and early endosomes in the *pan1-18TA* phosphorylation site mutant”).

*2) The authors failed to address the major concern that they are using 2D wide field microscopy to study endosomes, which move in three dimensions. Use of three-dimensional microscopy could improve the quality of the data presented in Figure 4 in particular. While I am disappointed that the authors still present only 2D microscopy, I believe that the data presented in Figure 4 are sufficient to support the authors' conclusions. A comment on this limitation is requested.*

In accordance with the reviewers’ suggestion, we have added sentences in the Results section (subsection “Pan1p-labeled endocytic vesicles associate with actin cables in the *pan1-18TA* mutant”).

*3) The layout of Figure 1, Figure 4 and Figure 6 should be changed such that the panels appear in alphabetical order. For example, panel F in Figure 6 seems to be out of place, coming after B and C but before D on the page.*

We modified those Figures to arrange all panels in alphabetical order.

*4) In the text (subsection “Interaction between endocytic vesicles and early endosomes in the* pan1-18TA *phosphorylation site mutant”), the authors claim that deletion of SLA2 leads to 95% reduction in actin clumps, but the representative image of a sla2Δ cell in Figure 6 clearly has an Abp1-labeled actin clump in the cytoplasm. In contrast, the myo3Δ/myo5Δ cell shown has no noticeable actin clump, even though the authors say in the text that clumps form more readily in this mutant. Two comments on this confusion: first, the authors should explicitly mention how they are quantifying internal clump formation; are the clumps 95% less bright in sla2Δ*

*cells? Or do they form 95% less frequently? Second, perhaps a better representative image could be chosen, such that the numbers stated in the text are borne out in the data shown in the figure.*

To mention how we quantified internal clump formation described in Figure 6—figure supplement 1, we have added a sentence in the legend of the Figure. We also changed the sentence “Deletion of Sla2p, a yeast Hip1R-related protein, in the *pan1-18TA* mutant decreased the formation of actin clumps by 95%~” to “Deletion of Sla2p, a yeast Hip1R-related protein, in the *pan1-18TA* mutant decreased the fraction of cells containing actin clumps by 95% ~”. We have also replaced the micrographs in Figure 6 (*sla2*△ and *myo3/5*△ cells) and Figure 6—figure supplement 1 (*myo3/5*△ cells) to better ones.

5) In the Discussion, first paragraph, the authors cite the vesicle formation defects in sla2 and ent1 actin-binding mutants as evidence that these proteins function before transport of endocytic vesicles on actin cables. However, predominant effects of the mutants on vesicle formation could obscure latter roles in vesicle transport and thus do not preclude such roles. Consequently, Sla2 and Ent1 could be responsible for the residual association of endocytic vesicles with actin cables in the Pan1-18TA actin-binding mutant. It might be worth editing the text in the Discussion to allow for this possibility.

According to the reviewers’ suggestion, we have changed the sentence from “~ these proteins seem to function at the steps before the transport of endocytic vesicles along the actin cable.” to “~ these proteins could be responsible for the residual association of endocytic vesicles with actin cables in the Pan1-18TA actin-binding mutant.”